# Private Hyperparameter Tuning with Ex-Post Guarantee

**Badih Ghazi**
Google Research
badihghazi@gmail.com

**Pritish Kamath**
Google Research
pritish@alum.mit.edu

**Alexander Knop**
Google Research
alexanderknop@google.com

**Ravi Kumar**
Google Research
ravi.k53@gmail.com

**Pasin Manurangsi**
Google Research
pasin@google.com

**Chiyuan Zhang**
Google Research
chiyuan@google.com

## Abstract

The conventional approach in differential privacy (DP) literature formulates the privacy-utility tradeoff with a "privacy-first" perspective: for a predetermined level of privacy, a certain utility is achievable. However, practitioners often operate under a "utility-first" paradigm, prioritizing a desired level of utility and then determining the corresponding privacy cost.

Wu et al. [2019] initiated a formal study of this "utility-first" perspective by introducing ex-post DP. They demonstrated that by adding correlated Laplace noise and progressively reducing it on demand, a sequence of increasingly accurate estimates of a private parameter can be generated, with the privacy cost attributed only to the least noisy iterate released. This led to a Laplace mechanism variant that achieves a specified utility with minimal privacy loss. However, their work, and similar findings by Whitehouse et al. [2022], are primarily limited to simple mechanisms based on Laplace or Gaussian noise.

In this paper, we significantly generalize these results. In particular, we extend the work of Wu et al. [2019] and Liu and Talwar [2019] to support any sequence of private estimators, incurring at most a doubling of the original privacy budget. Furthermore, we demonstrate that hyperparameter tuning for these estimators, including the selection of an optimal privacy budget, can be performed without additional privacy cost. Finally, we extend our results to ex-post Rényi DP, further broadening the applicability of utility-first privacy mechanisms.

## 1 Introduction

Many applications of machine learning and statistics involve computation on sensitive data, necessitating privacy-preserving techniques. In recent years, differential privacy (DP) [Dwork et al., 2016] has become one of the most rigorous formalization of privacy, with many practical applications [Abadi et al., 2016, Yu et al., 2024, Mehta et al., 2023, Tang et al., 2025, US Census Bureau, 2023, Hod and Canetti, 2025, Wilson et al., 2020]. Recall that an algorithm is DP if the output distributions on two neighboring inputs are close, where the closeness is determined by the *privacy budget*[1] $\varepsilon$:

**Definition 1** (Pure Differentially Privacy, [Dwork et al., 2016])**.** *For $\varepsilon \geq 0$, a mechanism $\mathcal{M}$ with input from $\mathcal{D}$ and output from $\mathcal{O}$ is $\varepsilon$-differentially private (or simply, $\varepsilon$-DP) iff $\Pr[\mathcal{M}(D) = o] \leq e^{\varepsilon} \Pr[\mathcal{M}(D') = o]$, for all $o \in \mathcal{O}$ and neighboring datasets $D, D' \in \mathcal{D}$.* [2]

---

[1]Our work also applies to *approximate-DP* with $\delta$ parameter; see Section 2 for the definition.

[2]We also assume for simplicity that $\mathcal{O}$ is finite; it is simple to extend the results to the infinite case.

39th Conference on Neural Information Processing Systems (NeurIPS 2025).

For reasons that will become clear soon, we refer to the classic DP definition above as *ex-ante* DP.

**Utility-First DP Mechanisms.** One of the main challenges in deploying DP is to ensure that the output remains useful. In particular, real-world deployments are often constrained by utility requirements. For example, in ML training, one may wish to ensure that the model accuracy meets a certain threshold. Similarly, in statistical applications, one may wish to guarantee that the *relative* error of the estimated population is small (e.g., [Ghazi et al., 2022]). Such desiderata may not be compatible with the ex-ante DP (Definition 1) since it *a priori* specifies a fixed privacy budget $\varepsilon$. This motivated Wu et al. [2019] to propose the notion of *ex-post* DP, where the privacy budget $\varepsilon$ can *depend on the output* of the mechanism, as formalized below.

**Definition 2** (Ex-post (Pure-) DP, [Wu et al., 2019]). *For a function $\tilde{\varepsilon} : \mathcal{O} \to \mathbb{R}^{\geq 0}$, a mechanism $\mathcal{M}$ with input from $\mathcal{D}$ and output from $\mathcal{O}$ is ex-post $\tilde{\varepsilon}$-DP iff $\Pr[\mathcal{M}(D) = o] \leq e^{\tilde{\varepsilon}(o)} \Pr[\mathcal{M}(D') = o]$, for all $o \in \mathcal{O}$ and neighboring datasets $D, D' \in \mathcal{D}$.*

Observe that any ex-post $\tilde{\varepsilon}$-DP mechanism is ex-ante $\varepsilon$-DP where $\varepsilon = \max_{o \in \mathcal{O}} \tilde{\varepsilon}(o)$. Furthermore, ex-post DP can also be used as a *privacy filter* to guarantee ex-ante DP [Rogers et al., 2023, Lebensold et al., 2024]. Roughly speaking, given a total privacy budget $\varepsilon$ for ex-ante DP, we run multiple ex-post algorithms where we subtract the realized privacy budget $\tilde{\varepsilon}(o)$ from $\varepsilon$ until the latter is exhausted.

Thus, the main question in ex-post DP becomes: What is the smallest privacy budget needed to produce an output that passes the desired utility bar? As pointed out in [Wu et al., 2019], a simple algorithm here is the "doubling" method where we start from a small privacy budget, run the "base" (ex-ante DP) algorithm with this budget, and continue until we find an acceptable output[3]. While simple, this doubling method can result in the privacy budget as large as four times[4] the optimal budget. Although there has been no improvement to this for general mechanisms, Wu et al. [2019] and later Whitehouse et al. [2022], building on an earlier work by Koufogiannis et al. [2016], gave an elegant improvement for the simple Laplace and Gaussian mechanisms that allows for a finer control of privacy budget increment than doubling and also just pays for the final privacy budget, instead of the total privacy budget (via composition). Alas, their method does not apply to more complex mechanisms, such as the seminal DP-SGD algorithm [Abadi et al., 2016] that is ubiquitous in private ML applications.

**Hyperparameter Tuning with DP.** A related challenge in private ML deployments is hyperparameter tuning. A naive solution here is to run any standard hyperparameter tuning algorithm and compute the total budget via composition theorems. However, this results in a prohibitive blow-up in the privacy budget, depending on the number of times the base algorithm is invoked. Liu and Talwar [2019] devised a simple algorithm but with a surprising guarantee. Their algorithm performs hyperparameter tuning on any ex-ante $\varepsilon$-DP by running it possibly multiple times (based on a carefully chosen distribution) and outputting the best found parameter. Even though the algorithm may be run many times, they show that the privacy budget incurred is only $3\varepsilon$. Furthermore, they show that, any "weakly useful" ex-ante DP hyperparameter tuning algorithm must incur privacy budget at least (roughly) $2\varepsilon$. A follow-up work by Papernot and Steinke [2022] closed this gap by giving an algorithm with privacy budget arbitrarily close to $2\varepsilon$, and further generalized this to work with Rényi DP [Mironov, 2017]. Although the task of optimizing the privacy budget in ex-post DP framework seems similar to hyperparameter tuning where $\varepsilon$ is a parameter, none of the aforementioned works [Liu and Talwar, 2019, Papernot and Steinke, 2022] applies to this setting since they require the base mechanism to have a fixed value of $\varepsilon$ in the ex-ante DP framework.

## 1.1 Our Contributions

In this work, we present the first hyperparameter tuning algorithm with ex-post DP guarantees. Our algorithm can take in multiple base mechanisms $\mathcal{M}_1, \ldots, \mathcal{M}_d$ where $\mathcal{M}_i$ is ex-ante $\varepsilon_i$-DP. It then runs these mechanisms (possibly multiple times, based on carefully crafted distributions) and select the "best" output. The ex-post DP guarantee is that, if the output comes from the base mechanism $\mathcal{M}_i$, then the privacy budget spent is only (roughly) $2\varepsilon_i$. We consider this *counterintuitive* and highly *surprising* given that other base mechanisms $\mathcal{M}_j$ with higher budget (i.e., $\varepsilon_j > \varepsilon_i$) might be run en

---

[3]Checking whether an output passes a utility bar must also be done with DP.

[4]A factor of two due to having to apply the composition theorem to sums up all the budget, and another factor of two from the potential misalignment between the doubling exponential grid and the optimal budget.

route and their output considered as part of the selection, nevertheless, our algorithm does not have to pay for this higher privacy budget $\varepsilon_j$! We are unaware of a similar phenomenon in DP.

While our algorithm (which works for multiple mechanisms with different $\varepsilon_i$'s) is a significant generalization of those of Liu and Talwar [2019], Papernot and Steinke [2022] (which only work for a single mechanism in the ex-ante setting), our privacy analysis is arguably simpler than theirs. In particular, the proof of our main privacy theorem (Theorem 8) draws inspiration from that of the Sparse Vector Technique [Dwork et al., 2009] and is elementary. We hope that the resulting simplicity will help further elucidate the underlining principles behind DP hyperparameter tuning. We note that our hyperparameter tuning is well suited for the task of optimizing the privacy budget given the privacy bar in ex-ante DP, since we can set the "score" in the selection step to be based on the privacy budget and whether the privacy bar is passed.

Finally, we introduce a notion of ex-post Rényi DP and show that hyperparameter tuning with Rényi DP is also possible (Theorem 10). In addition, we prove a connection between ex-post Rényi DP and ex-post approximate-DP and construct a privacy filter that allows composing together a sequence of ex-post Rényi DP mechanism into an ex-ante Rényi DP guarantee (which could allow using this algorithm in practical systems that want to provide ex-ante guarantees). Our technique, which applies to any mechanism including the aforementioned DP-SGD, is far more general than those in [Wu et al., 2019, Whitehouse et al., 2022], which only applies to Laplace or Gaussian mechanisms. To demonstrate this, we provide experiments that empirically show that our algorithm outperforms those in [Wu et al., 2019, Whitehouse et al., 2022] for linear regression (using the conversion from ex-post Rényi DP to ex-post approximate-DP).

## 2   Preliminaries

Let $\mathcal{D}$ be a set of datasets. We write $D \sim D'$ as a shorthand for a pair of neighboring input datasets (in $\mathcal{D}$). Let $\mathcal{O}$ be any set; for simplicity, we assume that $\mathcal{O}$ is discrete. We say that a function $\mathcal{M}$ mapping $D \in \mathcal{D}$ to a distribution over $\mathcal{O}$ is a *mechanism* with input from $\mathcal{D}$ and output from $\mathcal{O}$.

**Ex-Ante DP.**   While we have defined (ex-ante) *pure*-DP in Definition 1, it will be useful to recall other variants of DP. We start with approximate-DP, which allows an additional additive error $\delta$ in the difference in the two probabilities, as defined below. When $\delta = 0$, this coincides with Definition 1.

**Definition 3** (Differential Privacy, [Dwork et al., 2016])**.** *For $\varepsilon, \delta \geq 0$, a mechanism $\mathcal{M}$ with input from $\mathcal{D}$ and output from $\mathcal{O}$ is* ex-ante $(\varepsilon, \delta)$-differentially private *(or simply, $(\varepsilon, \delta)$-DP) iff* $\Pr[\mathcal{M}(D) \in S] \leq e^\varepsilon \Pr[\mathcal{M}(D') \in S] + \delta$, *for all $S \subseteq \mathcal{O}$ and all $D \sim D'$.*

Modern private learning is largely based on DP-SGD [Abadi et al., 2016].The privacy analysis of such a mechanism, which involves both subsampling and composition, is often done through Rényi DP [Mironov, 2017], which we recall here.

Let $\alpha > 1$, and $P$ and $Q$ be two distributions on $\mathcal{O}$. Let $D_\alpha(P \parallel Q)$ denote the *Rényi divergence* of $P$ from $Q$, i.e., $D_\alpha(P \parallel Q) = \frac{1}{\alpha-1} \log \sum_{o \in \mathcal{O}} (P(o))^\alpha (Q(o))^{1-\alpha}$.

**Definition 4** (Rényi DP, [Mironov, 2017])**.** *For $\alpha > 1$, $\varepsilon \geq 0$, a mechanism $\mathcal{M}$ with input from $\mathcal{D}$ and output from $\mathcal{O}$ is* ex-ante $(\alpha, \varepsilon)$-Rényi DP *(or simply, $(\alpha, \varepsilon)$-RDP) iff* $D_\alpha(\mathcal{M}(D) \parallel \mathcal{M}(D')) \leq \varepsilon$ *for all $D \sim D'$.*

**Ex-Post DP.**   We also need approximate and Rényi variants of ex-post DP. We start with the former since it is defined in the literature before this paper.

**Definition 5** (Ex-post DP, [Wu et al., 2019])**.** *For a function $\tilde{\varepsilon} : \mathcal{O} \to \mathbb{R}^{\geq 0}$ and $\delta > 0$, a mechanism $\mathcal{M}$ with input from $\mathcal{D}$ and output from $\mathcal{O}$ is* ex-post $(\tilde{\varepsilon}, \delta)$-DP *iff for all $S \subseteq \mathcal{O}$ and all $D \sim D'$,*

$$\sum_{o \in S} \Pr[\mathcal{M}(D) = o] \leq \sum_{o \in S} e^{\tilde{\varepsilon}(o)} \Pr[\mathcal{M}(D') = o] + \delta.$$

Again, when $\delta = 0$, this coincides with Definition 2. Next, we introduce ex-post Rényi DP.

**Definition 6** (Ex-post RDP)**.** *For a function $\varepsilon : \mathcal{O} \to \mathbb{R}^{\geq 0}$ and $\alpha > 1$, a mechanism $\mathcal{M}$ with input from $\mathcal{D}$ and output from $\mathcal{O}$ is* ex-post $(\alpha, \varepsilon)$-RDP *iff for all $D \sim D'$,*

$$\sum_{o \in \mathcal{O}} \frac{(\Pr[\mathcal{M}(D) = o])^\alpha}{(e^{\varepsilon(o)} \cdot \Pr[\mathcal{M}(D') = o])^{\alpha-1}} \leq 1.$$

We note that if $\tilde{\varepsilon}$ is a constant function, ex-ante and ex-post are equivalent for all DP notions stated.

In the case of ex-ante DP, it is known that ex-ante pure-DP is a stronger notion than ex-ante RDP, which in turn is stronger than ex-ante approximate-DP [Mironov, 2017]. We can show here that a similar result holds for their ex-post variants, as stated below. The proof, which follows its ex-ante counterpart, is deferred to the Supplementary Material.

**Lemma 7.** *Let $\tilde{\varepsilon} : \mathcal{O} \to \mathbb{R}^{\geq 0}$ be a function and $\alpha > 1$, $\delta \in [0, 1]$ be constants.*

- *If $\mathcal{M}$ is ex-post $\tilde{\varepsilon}$-DP, then $\mathcal{M}$ is ex-post $(\alpha, \tilde{\varepsilon})$-RDP.*
- *If $\mathcal{M}$ is ex-post $(\alpha, \tilde{\varepsilon})$-RDP, then $\mathcal{M}$ is ex-post $(\tilde{\varepsilon}', \delta)$-DP, where $\tilde{\varepsilon}'(o) = \tilde{\varepsilon}(o) + \frac{\log 1/\delta}{\alpha-1}$.*

**DP Selection Problem.**    The main focus of our paper is on the *DP selection* problem, which can be defined as follows. There are $d$ mechanisms $\mathcal{M}_1, \ldots, \mathcal{M}_d : \mathcal{D} \to \mathcal{O}$ where $\mathcal{M}_i$ is ex-ante DP (or ex-ante RDP). Following [Papernot and Steinke, 2022], we assume that $\mathcal{O}$ is a totally ordered set. The goal is to, after running $\mathcal{M}_1, \ldots, \mathcal{M}_d$ possibly multiple times, output $(o, i)$ where $i \in [d]$ and $o \in \mathcal{O}$ is an output from one of the runs of $\mathcal{M}_i$. Occasionally, we also allow an output $\perp$ to indicate that no good output was found.

A classic application of DP selection is in *DP hyperparameter tuning* of ML mechanisms. Here, each $\mathcal{M}_i$ can represent the mechanism with different configuration of parameters (including the privacy budgets) and the output $\mathcal{M}_i(D)$ is the private ML model together with the accuracy score on the test set.[5] Our setting also generalizes the widely-used exponential mechanism in which case $\mathcal{M}_i$ can be thought of as outputting the DP score of the $i$th candidate [McSherry and Talwar, 2007].

**Probability Notation.**    For a distribution $\mathcal{P}$, let $\mathrm{supp}(\mathcal{P})$ denote its support. For $i \in \mathrm{supp}(\mathcal{P})$, let $\mathcal{P}(i)$ denote the probability mass (resp., density) at $i$. For a subset $I$, let $\mathcal{P}(I) = \sum_{i \in I} \mathcal{P}(i)$ (resp., $\int_I \mathcal{P}(i)di$). Let $\mathrm{Ber}(p)$ denote the Bernoulli distribution with parameter $p$, i.e., the distribution on $\{0, 1\}$ such that the probability of $1$ equals $p$. Let $\mathrm{Geom}_p$ denote the geometric distribution with *failure probability* $p \in [0, 1]$, i.e., the distribution on $\mathbb{Z}_{\geq 0}$ such that $\mathrm{Geom}_p(k) = (1-p)p^k$. Throughout this work, we will use the following property of the Geometric distribution in our proofs:

$$\mathrm{Geom}_p(u) \leq p^{u-v} \cdot \mathrm{Geom}_p(v) \qquad\qquad \forall u, v \in \mathbb{Z} \text{ such that } u \leq v. \qquad (1)$$

Note that the above inequality is in fact an equality for $u \geq 0$.

Finally, let $\mathrm{Exp}_\lambda$ denote the exponential distribution with parameter $\lambda > 0$, i.e., the distribution on $\mathbb{R}_{>0}$ such that $\mathrm{Exp}_\lambda(x) = \lambda e^{-\lambda x}$.

# 3    Ex-Post Hyperparameter Tuning

In this section we present a new algorithm for hyperparameter tuning with ex-post DP guarantees (Algorithm 1). The main idea is *random dropping*, where we only include an output from each $\mathcal{M}_i$ to the candidate set $S$ with a certain probability. While this bears some similarity with the *random stopping* technique of Liu and Talwar [2019], our main innovation is the use of *correlated randomness* $k$ that is sampled at the beginning of the algorithm and determines the dropping probabilities of *all* the mechanisms. This idea is inspired by the Sparse Vector Technique (SVT) [Dwork et al., 2009], in which a threshold is noised at the beginning of the algorithm. Indeed, the high-level structure of our proof follows that of SVT: we couple $k$ with $k + 1$ in the two neighboring datasets, and bound the ratio of the output probabilities in the two cases. This is formalized in the proofs below. Furthermore, in Appendix A, we describe SVT (specifically, the AboveThreshold algorithm) and its connections to our method in more detail.

For convenience, we extend the order on $\mathcal{O}$ to $(\mathcal{O} \times [d]) \cup \{\perp\}$, where $\perp$ is the minimum element, and elements in $\mathcal{O} \times [d]$ are ordered lexicographically.

## 3.1    Pure-DP

Our privacy guarantee for pure-DP for Algorithm 1 is stated below. It says that, if the final output is from $\mathcal{M}_i$, then the privacy budget we pay is only $2\varepsilon_i + \varepsilon'$, where $\varepsilon'$ is a parameter of the distribution

---

[5]If the test set is considered sensitive, then we can add noise to achieve DP with respect to the test set.

---

**Algorithm 1** Hyperparameter Tuning Mechanism with Random Dropping.

---

**Parameters:** Distribution $\mathcal{E}$, Mechanisms $\mathcal{M}_i : \mathcal{D} \to \mathcal{O}$ and budget parameters $\varepsilon_i$ for $i \in [d]$
**Input:** Dataset $D$.
    $S \leftarrow \{\bot\}$
    Sample $k \sim \mathcal{E}$
    **for** $i = 1, \ldots, d$ **do**
        Sample $y_i \sim \mathrm{Ber}(e^{-\varepsilon_i \cdot k})$                                          {random drop}
        **if** $y_i = 1$ **then**
            $o \leftarrow \mathcal{M}_i(D_i)$
            $S \leftarrow S \cup \{(o, i)\}$
    **return** maximum element in $S$                        {as per the total order on $(\mathcal{O} \times [d]) \cup \{\bot\}$}

---

$\mathcal{E} = \mathrm{Geom}_{e^{-\varepsilon'}}$. This $\varepsilon'$ can be arbitrarily small, although setting it too small results in a larger drop probability. The latter can be mitigated by repeating each mechanism $\mathcal{M}_i$ multiple times in the input, which allows us to set that the desired expected number of times that each mechanism is run.

**Theorem 8** (Ex-post Pure-DP). *Let $\varepsilon' > 0$ and let each $\mathcal{M}_i$ be $\varepsilon_i$-DP. Define a function $\tilde{\varepsilon}$ such that $\tilde{\varepsilon}(o, i) = 2\varepsilon_i + \varepsilon'$ and $\tilde{\varepsilon}(\bot) = 0$. Then, Algorithm 1 with $\mathcal{E} = \mathrm{Geom}_{e^{-\varepsilon'}}$ is ex-post $\tilde{\varepsilon}$-DP.*

*Proof.* Consider neighboring datasets $D \sim D'$. Let $\mathcal{A}, \mathcal{A}'$ be the output distributions of Algorithm 1 on $D, D'$, respectively and let $\mathcal{Q}_i, \mathcal{Q}'_i$ be the output distributions of $\mathcal{M}_i$ on $D, D'$, respectively.

First, the probability that Algorithm 1 outputs $\bot$ is independent of input dataset and so $\mathcal{A}(\bot) = \mathcal{A}'(\bot)$. Next, consider any output $(o, i) \in \mathcal{O} \times [d]$. For each $j \in [d] \smallsetminus \{i\}$, let $U^j_{o,i} := \{o' \in \mathcal{O} \mid (o', j) > (o, i)\}$. Since $\mathcal{M}_j$ is $\varepsilon_j$-DP, it holds that $\mathcal{Q}_j(U^j_{o,i}) \geq e^{-\varepsilon_j} \mathcal{Q}'_j(U^j_{o,i})$. Similarly, we have $\mathcal{Q}_i(o) \leq e^{\varepsilon_i} \mathcal{Q}'_i(o)$. Finally, (1) yields $\mathrm{Geom}_{e^{-\varepsilon'}}(k) \leq e^{\varepsilon'} \cdot \mathrm{Geom}_{e^{-\varepsilon'}}(k+1)$. Thus, we have

$$
\mathcal{A}(o, i)
$$

$$
= \sum_{k=0}^{\infty} \mathrm{Geom}_{e^{-\varepsilon'}}(k) \cdot e^{-\varepsilon_i k} \mathcal{Q}_i(o) \cdot \prod_{j \neq i} \left(1 - e^{-\varepsilon_j k} \mathcal{Q}_j(U^j_{o,i})\right)
$$

$$
\leq \sum_{k=0}^{\infty} \left(e^{\varepsilon'} \cdot \mathrm{Geom}_{e^{-\varepsilon'}}(k+1)\right) \cdot e^{-\varepsilon_i k} \cdot (e^{\varepsilon_i} \mathcal{Q}'_i(o)) \cdot \prod_{j \neq i} \left(1 - e^{-\varepsilon_j k} \cdot \left(e^{-\varepsilon_j} \mathcal{Q}'_j(U^j_{o,i})\right)\right)
$$

$$
= e^{2\varepsilon_i + \varepsilon'} \sum_{k=0}^{\infty} \mathrm{Geom}_{e^{-\varepsilon'}}(k+1) \cdot e^{-\varepsilon_i(k+1)} \mathcal{Q}'_i(o) \cdot \prod_{j \neq i} \left(1 - e^{-\varepsilon_j(k+1)} \mathcal{Q}'_j(U^j_{o,i})\right)
$$

$$
\leq e^{2\varepsilon_i + \varepsilon'} \cdot \mathcal{A}'(o, i) \qquad \qquad \qquad \qquad \square
$$

The state-of-the-art (ex-ante) pure-DP hyperparameter tuning from [Papernot and Steinke, 2022, Corollary 3] can only take in a single mechanism $Q$ that is $\varepsilon$-DP. To compare this with our mechanism, consider the case where $\mathcal{M}_1 = \cdots = \mathcal{M}_d = Q$. In this setting, the two mechanisms are equivalent up to the difference in the distribution of the number of times $Q$ is executed. Figure 1 compares the standard deviation versus the mean of these two distributions. While our distribution has a larger variance, we emphasize its several advantages: our proof is completely elementary and our algorithm is more general as it works for different mechanisms with different $\varepsilon_i$'s values.

In addition, such a repetition trick allows us to prove a utility lower bound that achieves a "boosting" effect. To set a stage of the formal statement, note that we will give a relatively weak assumption that at least one of the mechanisms $\mathcal{M}_{i*}$ outputs a "good" candidate with a small probability $\alpha$. The theorem below states that, by repeating each mechanism $\mathcal{M}_i$ a certain number of times $T_i$, we can ensure that Algorithm 1 outputs a "good" candidate with probability at least $1 - \beta$ (where $\beta$ is a small number). We formalize this below, where "good" candidates are those that are at least $o^*$. Note also that $T_i$ is an upper bound on the number of times the mechanism $\mathcal{M}_i$ is run.[6]

---

[6]In ML settings, the score itself is computed as a measure of performance on a *test* set, as a proxy for the measure of performance on the *population distribution*. When running more mechanisms, one would need a

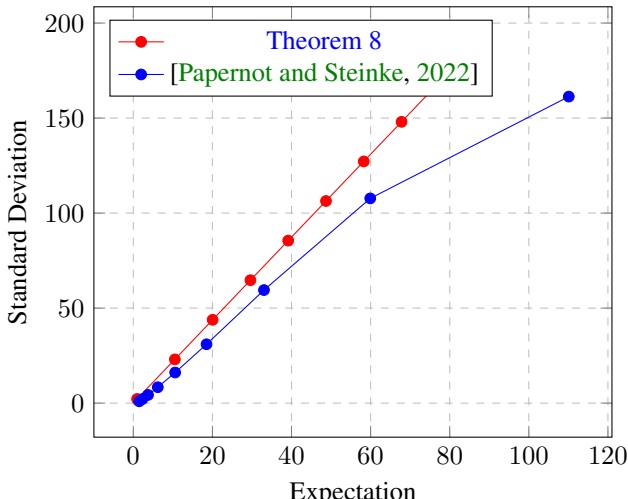

Figure 1: A plot of the standard deviation vs expectation of the number of invocations of a mechanism by the algorithms from Corollary 3 in [Papernot and Steinke, 2022] for $\eta = \varepsilon'/\varepsilon$ and Theorem 8 for $\mathcal{E} = \text{Geom}_{e^{-\varepsilon'}}$ for $\varepsilon = 0.1$ and $\varepsilon' = 0.01$.

**Theorem 9.** *Let $\alpha, \beta \in (0,1)$ and let $T_i = \left\lceil \frac{1}{\alpha} \left( \frac{2}{\beta} \right)^{\varepsilon_i/\varepsilon'} \cdot \ln \left( \frac{2}{\beta} \right) \right\rceil$. Consider Algorithm 1 with $\mathcal{E} = \text{Geom}_{e^{-\varepsilon'}}$ where, for all $i \in [d]$, we repeat $\mathcal{M}_i$ for $T_i$ times in the input parameter sequence. If there exists $o^* \in \mathcal{O}$ and $i^* \in [d]$ such that $\Pr[\mathcal{M}_{i^*}(D) \geq o^*] \geq \alpha$, then Algorithm 1 outputs an element that is larger than $(o^*, 0)$ with probability at least $1 - \beta$.*

*Proof.* If the final output is *smaller* than $(o^*, i^*)$, then in all runs of $\mathcal{M}_{i^*}$, it either has to be dropped or the output is less than $o^*$ (or both). For a fixed value of $k$, this happens with probability at most $(1 - e^{-\varepsilon_{i^*} \cdot k} \cdot \alpha)^{T_{i^*}} \leq \exp\left(-e^{-\varepsilon_{i^*} \cdot k} \cdot \alpha \cdot T_{i^*}\right)$. Due to our choice of $T_{i^*}$, this is at most $\beta/2$ for $k \leq \ln(2/\beta)/\varepsilon'$. Thus, the probability that the final output is smaller than $(o^*, i^*)$ is at most $\beta/2 + \Pr[k > \ln(2/\beta)/\varepsilon'] \leq \beta$. $\qquad\square$

Finally, we remark that, even in the ex-ante setting with all $\varepsilon$'s being equal, Liu and Talwar [2019] showed that the additional factor of 2 in the privacy budget is necessary even under a very weak assumption on the utility. This gives a strong evidence that our algorithm (in its generic form) requires such a factor of 2 blow-up as well.

## 3.2 Rényi DP

In this section, we consider the setting where each $\mathcal{M}_i$ satisfies Rényi DP (RDP) instead of pure-DP. As alluded to earlier, many popular DP machine learning algorithms, including DP-SGD [Abadi et al., 2016], do not satisfy pure-DP but are amenable to privacy analysis using RDP. By changing the distribution of $\mathcal{E}$ from the Geometric distribution (in the pure-DP case) to the Exponential distribution, we can show a version of Theorem 8 for RDP.

**Theorem 10** (Ex-post RDP). *Let $\ell_1, \ldots, \ell_d \geq 0$ and let us assume that each $\mathcal{M}_i$ is $(\alpha, \varepsilon_i)$-RDP with the output set $\mathcal{O} \times \mathbb{R}$. Let $\tau_i$ be the expected number of times $\mathcal{M}_i$ is executed in Algorithm 1 (note that it is independent of the dataset); and let $\tau = \sum_{i=1}^{d} \tau_i$.*

*Define a function $\tilde{\varepsilon}$ such that $\tilde{\varepsilon}(o, i) = (2 + \ell_i)\varepsilon_i + (1 + \ell_i)\varepsilon' + \frac{\log(\tau+1) + \sum_{j \neq i} e^{-\varepsilon_j(1 + \alpha\ell_i)}}{\alpha - 1}$ and $\tilde{\varepsilon}(\perp) = \frac{\log(\tau+1)}{\alpha - 1}$. Then, Algorithm 1 with $\mathcal{E} = \text{Exp}_{\varepsilon'}$ is $(\alpha, \tilde{\varepsilon})$-RDP.*

---

larger test set in order to get good generalization. This is orthogonal to Theorem 9, which in this setting would refer to $o^*$ as the performance on the population distribution.

We defer the proof of Theorem 10 to Appendix B, and provide a high-level overview here. Recall that in the above proof of Theorem 8 we use the inequality $\left(1 - e^{-\varepsilon_j k} \mathcal{Q}_j(U_{o,i}^j)\right) \leq \left(1 - e^{-\varepsilon_j(k+1)} \mathcal{Q}_j'(U_{o,i}^j)\right)$ which follows from the assumption that $\mathcal{M}_j$ is $\varepsilon_j$-DP. The main challenge in proving an RDP bound is that such an inequality fails since the assumption that $\mathcal{M}_j$ is $(\alpha, \varepsilon_j)$-RDP is weaker. To tackle this, we change the coupling: instead of coupling $k$ with $k+1$, we couple $k$ with $k+1+\ell_i$. Note that, since we allow $\ell_i$ to be non-integer (e.g., a value below one), this step necessitates the use of the Exponential distribution instead of the Geometric distribution. This new coupling allows us to instead compare $\left(1 - e^{-\varepsilon_j k} \mathcal{Q}_j(U_{o,i}^j)\right)$ with $\left(1 - e^{-\varepsilon_j(k+1+\ell_i)} \mathcal{Q}_j'(U_{o,i}^j)\right)$. Alas, the former is still not necessarily smaller than the latter. Nevertheless, via a careful argument, we can bound the ratio of these two quantities. Such a bound then ends up as the last term $\frac{\log(\tau+1) + \sum_{j \neq i} e^{-\varepsilon_j(1+\alpha \ell_i)}}{\alpha - 1}$ in our RDP guarantee in Theorem 10.

We note that, unlike Theorem 8, $\tilde{\varepsilon}(\perp) \neq 0$ in Theorem 10, i.e., we pay a privacy budget even when we fail to output anything meaningful. Again, this can be mitigated by repeating each mechanism multiple times in the input to decrease the probability of outputting $\perp$ to be arbitrarily small.

When the $\varepsilon_i$'s are different, it might be beneficial to pick $\ell_i$'s to be different as well. On the other hand, if we only consider the simple setting when $\varepsilon_1 = \cdots = \varepsilon_d = \varepsilon$ and we wish to choose $\ell_1, \ldots, \ell_d$ to all be equal to $\ell$. Then, it is not hard to verify that by setting $\ell = O\left(\frac{\log d}{\varepsilon \alpha}\right)$, we can ensure that $\sum_{j \in [d]} e^{-\varepsilon_j(1+\alpha \ell)} \leq 1$. With this setting of parameters and assuming $\varepsilon' \leq O(\varepsilon)$, we thus have the RDP bound of $\tilde{\varepsilon} = 2\varepsilon + \varepsilon' + O\left(\frac{\log d}{\alpha}\right)$. Note that this is similar to the bound from state-of-the-art (ex-ante) RDP hyperparameter tuning from [Papernot and Steinke, 2022, Theorem 2], which gives an RDP bound of $(2+\eta)\varepsilon + O\left(\frac{\log d}{\lambda}\right)$, where $\eta$ is the parameter of the negative binomial distribution (and assuming $\gamma \in (0,1)$ is a constant and $\hat{\lambda} = \lambda, \hat{\varepsilon} = \varepsilon$).

Alternatively, one may notice that $\tilde{\varepsilon}(o, i)$ doesn't depend on $\ell_j$ for $j \neq i$; hence, for each $i$ it is possible to choose $\ell_i$ as a value minimizing $\tilde{\varepsilon}(o, i)$.

## 4  Fully-Adaptive Composition with Ex-Post Rényi DP

Real-life applications of DP mechanism are often highly interactive: i.e., the analyst queries private data and based on the results of these queries decides what to query next. Moreover, often it is important to be able to choose further privacy parameters based on previous responses. Following Rogers et al. [2016], we express this interactivity in a form of a "game" between an adversary $\mathcal{A}$ and some system $\mathcal{F}_{\alpha, \varepsilon}$. In this interaction there is an unknown bit that the adversary wishes to learn; on each step $i$ the adversary (based on previous responses) chooses two datasets $D_i^{(0)}$ and $D_i^{(1)}$, a privacy loss function $\tilde{\varepsilon}_i$, and a mechanism $\mathcal{M}_i$ that is $(\alpha, \tilde{\varepsilon}_i)$-RDP; the system decides if such request could be answered; and if the system allows to proceed, the result $\mathcal{M}_i(D_i^{(b)})$ is given to the adversary. Our *privacy filter* is simple: Start with a total RDP budget $\varepsilon$, subtract from it the ex-post RDP bound after each request is answered, and only allow the next request to be answered if the remaining budget is at least the maximum possible ex-post RDP bound of the mechanism. See Algorithm 2 for the details.

---

**Algorithm 2** Privacy filter for ex-post RDP.

---

**Parameters:** Order $\alpha > 1$, privacy budget $\varepsilon > 0$, number of steps $n$.
**Input:** Adversary $\mathcal{A}$, private bit $b \in \{0, 1\}$.
    **for** $i$ from 1 to $n$ **do**
        $D_i^{(0)}, D_i^{(1)}, \tilde{\varepsilon}_i, \mathcal{M}_i \leftarrow \mathcal{A}(o_1, \ldots, o_{i-1})$
        **if** $\sum_{j=1}^{i-1} \tilde{\varepsilon}_j(o_j) + \sup_o \tilde{\varepsilon}_i(o) > \varepsilon$ **then**
            **return** $o_1, \ldots, o_{i-1}$
        $o_i \leftarrow \mathcal{M}_i(D_i^{(b)})$
    **return** $o_1, \ldots, o_n$

---

Our privacy filter allows us to use ex-post RDP algorithms in interactive manners while ensuring a final ex-ante RDP bound. This result extends the results of Lécuyer [2021], Feldman and Zrnic [2021] to allow adversary to issue mechanisms with ex-post guarantees. We note that such a connection between ex-post DP and ex-ante DP via a privacy filter has been made before, e.g., for pure-DP and approximate-DP [Rogers et al., 2016, Lebensold et al., 2024], and for specific RDP mechanisms like Brownian Noise Reduction [Rogers et al., 2023]. We believe our work is the first to generalize this filter to the full, arbitrary class of ex-post RDP mechanisms, although the proof of our filter follows simply from the aforementioned previous work. We defer the full proof to Appendix C.

**Theorem 11.** *For any adversary $\mathcal{A}$, $\alpha > 1$, $\varepsilon > 0$, $n \in \mathbb{N}$, $\mathrm{D}_\alpha \left( \mathrm{IT}^0(\mathcal{F}_{\alpha,\varepsilon}; \mathcal{A}) \, \| \, \mathrm{IT}^1(\mathcal{F}_{\alpha,\varepsilon}; \mathcal{A}) \right) \leq \varepsilon$, where $\mathrm{IT}^b(\mathcal{F}_{\alpha,\varepsilon}; \mathcal{A})$ is the output of Algorithm 2.*

# 5  Experiments

We present two sets of experiments: In the first, we evaluate the performance of our algorithm on analytical tasks and in the second, we focus on the performance on a machine learning problem.

## 5.1  Analytical Problem

Informally the problem is as follows [Rogers et al., 2023]: given a message board, the goal is to estimate the number of unique users per thread, each with relative error 10%; we want as many estimates as possible. Here, a user could contribute to any of the threads. We consider two datasets.

**Synthetic:** The synthetic datasets are generated as follows: $N \in \{8000, 16000, 32000, 64000, 128000\}$ samples are obtained from the power-law distribution with support on $[300]$ (i.e., the distribution such that for $x \in [300]$, the density is proportional to $x^{0.75}$ and is 0 otherwise). We assume that each $x$ corresponds to a thread and the number of samples with this value is the number of users. Hence, we convert these samples into a histogram of 300 values with their counts.

**Reddit:** We use the `webis/tldr-17` dataset [Völske et al., 2017] that contains authors of posts and subreddits where the post was posted. The histogram consists of subreddits (i.e., threads) and the number of unique users who posted in the subreddit.

We consider two types of algorithms: one where a pure-DP guarantee is available and another where we eventually have an approximate-DP guarantee. However, in both cases, we can check whether the current estimate $\hat{y}$ is good (i.e., we expect it to be with less than 10% error) by checking that $|(\hat{y} + \sigma)/(\hat{y} - \sigma)| \in [0.9, 1.1]$ and $|\hat{y}| \geq \sigma$, where $\sigma$ is the standard deviation of the noise used to obtain the estimate.

For pure-DP, we follow [Rogers et al., 2023] and allow mechanisms to compute each estimate with privacy budget $\varepsilon = 0.001 \cdot (\sqrt{2})^i$ for some $i$, with a total budget of 10. The comparison includes the doubling mechanism with Laplace noise (the algorithm that attempts one $\varepsilon$ after another and pays for them via composition) [Wu et al., 2019], noise reduction method with Laplace noise from [Wu et al., 2019], and Algorithm 1 with Laplace mechanism and $\varepsilon' = 0.001$. The detailed results can be seen in Table 1. Note that in terms of number of produced answers, our algorithm outperforms all other solutions and in terms of precision (percentage of outputs that were indeed with 10% relative error) is similar to the doubling estimator and within reasonable bounds.

For approximate-DP, we allow mechanisms to compute each estimate with privacy budget $\varepsilon = 0.001 \cdot (\sqrt{2})^i$ for some $i$, with a total budget of $(10, 10^{-6})$. We compare the doubling mechanism with Gaussian noise and zCDP budgeting [Bun and Steinke, 2016], the Brownian Motion algorithm with zCDP budgeting [Whitehouse et al., 2022], and Algorithm 1 with Gaussian mechanism and RDP budgeting. The results can be seen in Table 2. In this case, our algorithm underperforms, which is not too surprising since the Gaussian mechanism with zCDP budgeting is tailored for tasks of this nature.

Table 2: Comparison between approximate-DP mechanisms on synthetic data; the column 'Produced Answers' contains the average and standard deviation of the number of threads that the algorithm was able to estimate before the budget got exhausted and the column 'Precision' contains the average and standard deviation of the fraction of threads that were estimated with less than 10% relative error among the estimated columns. A cell value $a_{\pm b}$ means $a$ is the average and $b$ is the standard deviation. The dataset name S$N$ means synthetic dataset made of $N$ samples.

| Dataset | Brownian Motion Mechanism | | Doubling Mechanism | | Algorithm 1 w/ Gaussian | |
| --- | --- | --- | --- | --- | --- | --- |
| | Precision | Produced Answers | Precision | Produced Answers | Precision | Produced Answers |
| S8000 | $0.974_{\pm 0.03}$ | $28.63_{\pm 0.74}$ | $0.969_{\pm 0.05}$ | $17.20_{\pm 0.57}$ | $0.970_{\pm 0.09}$ | $6.694_{\pm 0.46}$ |
| S16000 | $0.973_{\pm 0.02}$ | $50.44_{\pm 0.89}$ | $0.971_{\pm 0.03}$ | $30.75_{\pm 0.65}$ | $0.972_{\pm 0.05}$ | $11.97_{\pm 0.33}$ |
| S32000 | $0.974_{\pm 0.02}$ | $88.58_{\pm 1.01}$ | $0.973_{\pm 0.02}$ | $54.74_{\pm 0.80}$ | $0.971_{\pm 0.04}$ | $20.61_{\pm 0.51}$ |
| S64000 | $0.975_{\pm 0.01}$ | $154.8_{\pm 1.29}$ | $0.974_{\pm 0.02}$ | $96.95_{\pm 1.01}$ | $0.973_{\pm 0.03}$ | $33.84_{\pm 0.63}$ |
| S128000 | $0.977_{\pm 0.01}$ | $269.7_{\pm 1.50}$ | $0.980_{\pm 0.01}$ | $173.6_{\pm 1.23}$ | $0.970_{\pm 0.03}$ | $52.33_{\pm 1.17}$ |

Table 1: Comparison between pure-DP mechanisms; the column 'Produced Answers' contains the average and standard deviation of the number of threads that the algorithm was able to estimate before the budget got exhausted and the column 'Precision' contains the average and standard deviation of the fraction of threads that were estimated with less than 10% relative error among the estimated columns. A cell value $a_{\pm b}$ means $a$ is the average and $b$ is the standard deviation. The dataset name S$N$ means synthetic dataset made of $N$ samples.

| Dataset | Doubling Mechanism | | Noise Reduction Mechanism | | Algorithm 1 w/ Laplace | |
| --- | --- | --- | --- | --- | --- | --- |
| | Precision | Produced Answers | Precision | Produced Answers | Precision | Produced Answers |
| S8000 | $0.911_{\pm 0.07}$ | $14.77_{\pm 0.47}$ | $0.999_{\pm 0.02}$ | $2.22_{\pm 1.45}$ | $0.912_{\pm 0.06}$ | $20.37_{\pm 0.52}$ |
| S16000 | $0.912_{\pm 0.06}$ | $22.47_{\pm 0.54}$ | $0.999_{\pm 0.02}$ | $4.47_{\pm 2.22}$ | $0.911_{\pm 0.05}$ | $30.63_{\pm 0.57}$ |
| S32000 | $0.910_{\pm 0.05}$ | $33.96_{\pm 0.56}$ | $0.998_{\pm 0.12}$ | $8.12_{\pm 3.29}$ | $0.905_{\pm 0.04}$ | $45.74_{\pm 0.63}$ |
| S64000 | $0.909_{\pm 0.04}$ | $50.90_{\pm 0.61}$ | $0.998_{\pm 1.72}$ | $15.14_{\pm 5.25}$ | $0.911_{\pm 0.03}$ | $68.39_{\pm 0.73}$ |
| S128000 | $0.909_{\pm 0.03}$ | $76.09_{\pm 0.74}$ | $0.997_{\pm 0.01}$ | $27.68_{\pm 9.15}$ | $0.912_{\pm 0.03}$ | $102.1_{\pm 0.88}$ |
| Reddit | $0.911_{\pm 0.02}$ | $279.7_{\pm 1.10}$ | $0.992_{\pm 0.01}$ | $207.2_{\pm 42.1}$ | $0.922_{\pm 0.01}$ | $327.5_{\pm 13.9}$ |

## 5.2 Machine Learning

We perform the following experiments related to an ML task.

1. In the first set of experiments we follow the setup from [Wu et al., 2019, Whitehouse et al., 2022] and train a linear regression model on a dataset of timeseries generated by Twitter usage [The AMA Team at Laboratoire d'Informatique de Grenoble] (subsampled to 100000 data-points) and search for a model with at most 0.05 MSE. We compare the following mechanisms.

   (a) Brownian motion with the AboveThreshold mechanism using sufficient statistics perturbation [Vu and Slavkovic, 2009], a sequence 0.1, 0.2, ... 1 of values of $\varepsilon$ for Brownian motion, and 0.01 for AboveThreshold on the MSE of the model.

   (b) Algorithm 1 with the DP-SGD [Abadi et al., 2016] mechanism, learning linear models with $\varepsilon' = 0.01$, possible values of $\varepsilon$ in $\{0.1, 0.2, \dots 1\}$, learning rate in $\{0.01, 0.1, 1\}$, epochs in $\{1, 5, 10\}$, batch sizes in $\{32, 64, 128, 256, 512, 1000\}$, and clipping norms in $\{0.1, 1, 10\}$.

   (c) Doubling mechanism Wu et al. [2019] running DP-SGD tuned according to Papernot and Steinke [2022] with identical hyperparameters to those used by Algorithm 1.

2. In the second set, we train a classifier for the MNIST dataset [LeCun et al., 2010] and search for the minimal $\varepsilon$ such that the model has at least 0.6 accuracy. We compare the following mechanisms.

Table 3: Comparison of the $\varepsilon$ values used by the Brownian Motion mechanism, doubling mechanism, and Algorithm 1 when applied to machine learning tasks. The numbers represent the average ex-post $(\varepsilon, 10^{-6})$-DP guarantees over 100 trials.

| Dataset | Brownian Motion | Doubling Mechanism | Algorithm 1 |
|---------|-----------------|--------------------|-------------|
| Twitter | 0.77 | 0.55 | 0.28 |
| MNIST | 0.62 | 0.38 | 0.32 |
| Gisette | 0.33 | 0.54 | 0.23 |

(a) Brownian motion with the AboveThreshold mechanism using output perturbation [Vu and Slavkovic, 2009], a sequence $0.1, 0.2, \ldots 1$ of values of $\varepsilon$ for Brownian motion, and $0.01$ for AboveThreshold on accuracy of the model.

(b) Algorithm 1 with DP-SGD mechanisms learning CNN models (for the architecture see Basic MNIST Example) with $\varepsilon' = 0.01$, possible values of $\varepsilon$ in $\{0.1, 0.2, \ldots 1\}$, learning rate in $\{0.01, 0.1, 1\}$, epochs in $\{1, 5, 10\}$, batch sizes in $\{32, 64, 128, 256, 512, 1000\}$, and clipping norms in $\{0.1, 1, 10\}$.

(c) Doubling mechanism Wu et al. [2019] running DP-SGD tuned according to Papernot and Steinke [2022] with identical hyperparameters to those used by Algorithm 1.

3. In the third set, we train a classifier for the Gisette [Guyon et al., 2004] dataset and search for the minimal $\varepsilon$ such that the model has at least $0.4$ accuracy. We compare the following mechanisms.

(a) Brownian motion with the AboveThreshold mechanism using output perturbation [Vu and Slavkovic, 2009], a sequence $0.1, 0.2, \ldots 1$ of values of $\varepsilon$ for Brownian motion, and $0.01$ for AboveThreshold on accuracy of the model.

(b) Algorithm 1 with DP-SGD mechanisms learning a linear model with $\varepsilon' = 0.01$, possible values of $\varepsilon$ in $\{0.1, 0.2, \ldots 1\}$, learning rate in $\{0.01, 0.1, 1\}$, epochs in $\{1, 5, 10\}$, batch sizes in $\{32, 64, 128, 256, 512, 1000\}$, and clipping norms in $\{0.1, 1, 10\}$.

(c) Doubling mechanism Wu et al. [2019] running DP-SGD tuned according to Papernot and Steinke [2022] with identical hyperparameters to those used by Algorithm 1.

(In both cases we use Opacus [Yousefpour et al., 2021] for training DP-SGD.)

The results of comparison can be seen in Table 3. Our algorithm significantly outperforms the previous Brownian motion algorithms and doubling mechanism. This can be explained by the fact that DP-SGD vastly outperforms the simpler models in these settings [Yu et al., 2020] and the fact that doubling requires running tuning which (in order to keep the budget small) needs high $\alpha$. Our algorithm also consistently outperforms the doubling mechanism. This superior performance can be attributed to the doubling mechanism's privacy loss being approximately two times greater than that of the tuning mechanism which in-turn is about two times greater than the underlying procedure.

## 6    Conclusion and Open Problems

In this work, we give a simple yet general algorithm for DP hyperparameter tuning that works even for ex-post DP and RDP. Despite its generality, our experiments show that it achieves significant advantage over previous algorithms for ML applications. Two immediate questions remain. First, is it possible to get rid of the $\ell$'s in Theorem 10? Second, and somewhat related, is the question of proving a zCDP version of the result, which would improve the analysis in the case of analytics workloads since the Gaussian mechanism is typically used in those cases.

**Acknowledgments**

We thank the anonymous reviewers for their valuable feedback. We also thank Ryan Rogers for helpful discussion and for clarifying the connection between their work on privacy filters [Rogers et al., 2023] and the ex-post RDP framework presented in this paper.

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

# A Warm-Up: Ex-Post AboveThreshold Mechanisms

Before we present our full ex-post DP Hyperparameter Tuning algorithm, it would be helpful to recall the Sparse Vector Technique [Dwork et al., 2009]. In particular, our ex-post DP Hyperparameter Tuning algorithm derives inspiration from the so-called AboveThreshold mechanism.

## A.1 Classic AboveThreshold Mechanism

To state the AboveThreshold mechanism, recall that the sensitivity of a function $f : \mathcal{D} \to \mathbb{Z}$ is defined as $\Delta(f) := \max_{D \sim D'} |f(D) - f(D')|$, where the maximum is taken over all neighboring input datasets $D \sim D'$. The setting here is that we are given sensitivity-1 functions $f_1, \ldots, f_d$ and the goal is to output the first index $i$ such that $f_i(D)$ is at least zero.[7] The mechanism works by first sampling a Geometric noise $k$ to be its noisy threshold; then, for each $f_i$, we add an independent Geometric noise $y_i$ to it and check if it exceeds the threshold. If it does, we output $i$ and terminate. It turns out that, in addition to $i$, we can get an estimate of $f_i$ (via $f_i(D) + y_i - k$) for free without any additional privacy cost [Ding et al., 2023]. A full description is given in Algorithm 3.

---

**Algorithm 3** AboveThreshold Mechanism

---

**Parameters:** Sensitivity-1 functions $f_i : \mathcal{D} \to \mathbb{Z}$ and budget parameters $\varepsilon_i$ for $i \in [d]$, and additional privacy budget $\varepsilon' > 0$.
**Input:** Dataset $D$.
  Sample $k \sim \mathrm{Geom}_{e^{-\varepsilon'}}$                                             {threshold noise}
  **for** $i = 1, \ldots, d$ **do**
    Sample $y_i \sim \mathrm{Geom}_{e^{-\varepsilon_i}}$                                        {query noise}
    **if** $f_i(D) + y_i \geq k$ **then**
      **return** $(f_i(D) + y_i - k, i)$ and terminate
  **return** $\perp$

---

This algorithm generalizes the standard ex-ante DP AboveThreshold mechanism since we allow the noise for each $f_i$ to have different privacy budget parameter $\varepsilon_i$. Indeed, with this mechanism, we show that the ex-post privacy budget spent for releasing $f_i$ is only $2\varepsilon_i + \varepsilon'$, as stated below.

**Theorem 12** (Ex-post AboveThreshold). *Define a function $\tilde{\varepsilon}$ such that $\tilde{\varepsilon}(o, i) = 2\varepsilon_i + \varepsilon'$ for all $o \in \mathbb{Z}_{\geq 0}, i \in [d]$ and $\tilde{\varepsilon}(\perp) = \varepsilon'$. Then, Algorithm 3 is ex-post $\tilde{\varepsilon}$-DP.*

Our proof closely mirrors the proof for the analogous ex-ante AboveThreshold. Namely, for neighboring datasets $D, D'$ and output $(o, i)$, we can couple the Geometric noises such that $k' = k + 1, y_i' = y_i + 1 + f_i(D) - f_i(D')$ and all other noises remain the same. It is not hard to see that, if the algorithm returns $(o, i)$ on $D$, it returns $(o, i)$ on $D'$ as well. Furthermore, due to property (1) of the Geometric distribution, the probability decreases by at most $e^{2\varepsilon_i + \varepsilon'}$ factor. This idea is formalized in the proof given below.

*Proof of Theorem 12.* Consider neighboring datasets $D \sim D'$. Let $\mathcal{A}, \mathcal{A}'$ be the output distributions of Algorithm 3 on $D, D'$, respectively. Below, we write $\mathrm{Geom}_p(< x)$ as a shorthand for $\mathrm{Geom}_p(\{x - 1, x - 2, \ldots\}) = \sum_{y=0}^{x-1} \mathrm{Geom}_p(y)$.

First, consider any output $(o, i)$. This output happens exactly when $y_i = o + k - f_i(D)$ and $y_j < k - f_j(D)$ for all $j < i$. Thus, we have

$$\mathcal{A}(o, i) = \sum_{k=0}^{\infty} \mathrm{Geom}_{e^{-\varepsilon'}}(k) \cdot \mathrm{Geom}_{e^{-\varepsilon_i}}(o + k - f_i(D)) \prod_{j=1}^{i-1} \mathrm{Geom}_{e^{-\varepsilon_j}}(< k - f_j(D)) \quad (2)$$

Since $\Delta(f_i) \leq 1$, we have $o + k - f_i(D) \leq o + k + 1 - f_i(D')$; applying (1) then yields

$$\mathrm{Geom}_{e^{-\varepsilon_i}}(o + k - f_i(D)) \leq e^{\varepsilon_i \cdot (1 - f_i(D') + f_i(D))} \cdot \mathrm{Geom}_{e^{-\varepsilon_i}}(o + k + 1 - f_i(D'))$$

$$\leq e^{2\varepsilon_i} \cdot \mathrm{Geom}_{e^{-\varepsilon_i}}(o + k + 1 - f_i(D')), \quad (3)$$

---

[7]We can easily handle non-zero threshold $\tau$ by considering $f_i - \tau$ instead.

where the second inequality again uses $\Delta(f_i) \leq 1$.

Furthermore, $\Delta(f_j) \leq 1$ implies $\mathrm{Geom}_{e^{-\varepsilon_j}}(< k - f_j(D)) \leq \mathrm{Geom}_{e^{-\varepsilon_j}}(< k + 1 - f_j(D'))$. Moreover, (1) implies $\mathrm{Geom}_{e^{-\varepsilon'}}(k) \leq e^{\varepsilon'} \cdot \mathrm{Geom}_{e^{-\varepsilon'}}(k+1)$. Plugging these two inequalities and (3) into (2) yields

$$\mathcal{A}(o, i)$$

$$\leq \sum_{k=0}^{\infty} e^{\varepsilon'} \cdot \mathrm{Geom}_{e^{-\varepsilon'}}(k+1) \cdot e^{2\varepsilon_i} \cdot \mathrm{Geom}_{e^{-\varepsilon_i}}(o+k+1-f_i(D')) \cdot \prod_{j=1}^{i-1} \mathrm{Geom}_{e^{-\varepsilon_j}}(< k+1-f_j(D'))$$

$$= e^{2\varepsilon_i+\varepsilon'} \sum_{k=0}^{\infty} \mathrm{Geom}_{e^{-\varepsilon'}}(k+1) \cdot \mathrm{Geom}_{e^{-\varepsilon_i}}(o+k+1-f_i(D')) \cdot \prod_{j=1}^{i-1} \mathrm{Geom}_{e^{-\varepsilon_j}}(< k+1-f_j(D'))$$

$$\leq e^{2\varepsilon_i+\varepsilon'} \cdot \mathcal{A}'(o, i).$$

Next, consider the output $\perp$. This output happens when $y_j < k - f_j(D)$ for all $j \in [d]$. Thus, we have

$$\mathcal{A}(\perp) = \sum_{k=0}^{\infty} \mathrm{Geom}_{e^{-\varepsilon'}}(k) \cdot \prod_{j \in [d]} \mathrm{Geom}_{e^{-\varepsilon_j}}(< k - f_j(D))$$

$$\leq \sum_{k=0}^{\infty} \mathrm{Geom}_{e^{-\varepsilon'}}(k) \cdot \prod_{j \in [d]} \mathrm{Geom}_{e^{-\varepsilon_j}}(< k+1 - f_j(D'))$$

$$\leq \sum_{k=0}^{\infty} e^{\varepsilon'} \cdot \mathrm{Geom}_{e^{-\varepsilon'}}(k+1) \cdot \prod_{j \in [d]} \mathrm{Geom}_{e^{-\varepsilon_j}}(< k+1 - f_j(D'))$$

$$= e^{\varepsilon'} \sum_{k=0}^{\infty} \mathrm{Geom}_{e^{-\varepsilon'}}(k+1) \cdot \prod_{j \in [d]} \mathrm{Geom}_{e^{-\varepsilon_j}}(< k+1 - f_j(D'))$$

$$\leq e^{\varepsilon'} \cdot \mathcal{A}'(\perp),$$

where again we use $\Delta(f) \leq 1$ in the first inequality and (1) in the subsequent inequality. $\square$

### A.2 Optimized Noise via Monotonicity

Let $\succeq$ denote any total order on $\mathcal{D}$. We say that a function $f$ is monotone (with respect to $\succeq, \sim$) iff the following holds: $f(D) \geq f(D')$ for all $D \sim D'$ such that $D \succeq D'$. An example of this is when $\sim$ denotes an add-remove neighboring notion, i.e., $D \sim D'$ iff $D$ results from adding or removing a user from $D'$; in this case, we may let $\succeq$ be based on the size of the dataset, and $f$ is monotone iff adding a user does not decrease the function value. Such a property holds when $f$ is counting the number of users satisfying certain criteria, which is an example used in our experiment in Section 5.

For monotone $f$, the same algorithm (Algorithm 3) yields a better ex-post guarantee, where we do not need to pay the factor of 2 in front of $\varepsilon_i$, as stated below. Note that this is similar to a saving seen in the ex-ante setting Ding et al. [2023].

**Theorem 13** (Ex-post Monotone AboveThreshold). *Define a function $\tilde{\varepsilon}$ such that $\tilde{\varepsilon}(i) = \varepsilon_i + \varepsilon'$ and $\tilde{\varepsilon}(\perp) = \varepsilon'$. If $f$ is monotone, then Algorithm 3 is ex-post $\tilde{\varepsilon}$-DP.*

The proof proceeds similarly to before except that, in the monotone case, either (i) $f_i(D') \geq f_i(D)$ in which case the difference $y'_i - y_i$ is already at most one (instead of two as before), or (ii) $f_i(D') \leq f_i(D)$ in which case we can instead couple with $k' = k, y'_i = y_i + f_i(D) - f_i(D')$ resulting in $y'_i - y_i \leq 1$ again.

*Proof of Theorem 13.* We use similar notations as in the proof of Theorem 12. The case of $\perp$ output is exactly the same as in that proof. For the output $(o, i)$, we consider the following two subcases, based on whether $D' \succeq D$. First, let us consider the case $D' \succeq D$. In this case, the proof is exactly the same as before except that, since $f_i(D') \geq f_i(D)$, in (3), we instead get

$$\mathrm{Geom}_{e^{-\varepsilon_i}}(o+k-f_i(D)) \leq e^{\varepsilon_i \cdot (1-f_i(D')+f_i(D))} \cdot \mathrm{Geom}_{e^{-\varepsilon_i}}(o+k+1-f_i(D'))$$

$$\leq e^{\varepsilon_i} \cdot \mathrm{Geom}_{e^{-\varepsilon_i}}(o + k + 1 - f_i(D')).$$

Following the same line of reasoning as before, we then get $\mathcal{A}(o, i) \leq e^{\varepsilon_i + \varepsilon'} \cdot \mathcal{A}'(o, i)$ as desired.

Finally, let us consider the case $D \succeq D'$. In this case, since $f_j(D) \geq f_j(D')$, we have

$$\mathcal{A}(o, i) = \sum_{k=0}^{\infty} \mathrm{Geom}_{e^{-\varepsilon'}}(k) \cdot \mathrm{Geom}_{e^{-\varepsilon_i}}(o + k - f_i(D)) \prod_{j=1}^{i-1} \mathrm{Geom}_{e^{-\varepsilon_j}}(< k - f_j(D))$$

$$\leq \sum_{k=0}^{\infty} \mathrm{Geom}_{e^{-\varepsilon'}}(k) \cdot \mathrm{Geom}_{e^{-\varepsilon_i}}(o + k - f_i(D)) \prod_{j=1}^{i-1} \mathrm{Geom}_{e^{-\varepsilon_j}}(< k - f_j(D'))$$

Since $o + k - f_i(D) \leq o + k - f_i(D')$, we can apply (1) to arrive at

$$\mathrm{Geom}_{e^{-\varepsilon_i}}(o + k - f_i(D)) \leq e^{\varepsilon_i \cdot (f_i(D) - f_i(D'))} \cdot \mathrm{Geom}_{e^{-\varepsilon_i}}(o + k - f_i(D'))$$

$$\leq e^{\varepsilon_i} \cdot \mathrm{Geom}_{e^{-\varepsilon_i}}(o + k - f_i(D')),$$

where the second inequality follows from $\Delta(f_i) \leq 1$.

Combining the above two inequalities then gives

$$\mathcal{A}(o, i) \leq \sum_{k=0}^{\infty} \mathrm{Geom}_{e^{-\varepsilon'}}(k) \cdot \mathrm{Geom}_{e^{-\varepsilon_i}}(o + k - f_i(D)) \prod_{j=1}^{i-1} \mathrm{Geom}_{e^{-\varepsilon_j}}(< k - f_j(D'))$$

$$\leq \sum_{k=0}^{\infty} \mathrm{Geom}_{e^{-\varepsilon'}}(k) \cdot e^{\varepsilon_i} \cdot \mathrm{Geom}_{e^{-\varepsilon_i}}(o + k - f_i(D')) \prod_{j=1}^{i-1} \mathrm{Geom}_{e^{-\varepsilon_j}}(< k - f_j(D'))$$

$$= e^{\varepsilon_i} \cdot \mathcal{A}'(o, i)$$

$$\leq e^{\varepsilon_i + \varepsilon'} \cdot \mathcal{A}'(o, i),$$

which concludes our proof. $\qquad\square$

### A.3 Generalized AboveThreshold Mechanism via Random Dropping

Next, we present a new generalized algorithm for AboveThreshold with ex-post DP guarantees (Algorithm 4). We consider the following general setting: We have mechanisms $M_1, \ldots, M_d : \mathcal{D} \to \mathcal{O}$ and the goal is to output the first mechanism such that $M_i(D)$ is at least a certain threshold $\tau \in \mathcal{O}$. Our main idea is *random dropping*, where, instead of always comparing $M_i(D)$ with $\tau$, we only compare with a certain probability; otherwise, we drop $M_i(D)$ completely. While this bears some similarity with the *random stopping* technique of Liu and Talwar [2019], our main innovation is the use of *correlated randomness* $k$ that is sampled at the beginning of the algorithm and determines the dropping probabilities of *all* the mechanisms. This idea is inspired by the above analysis of the AboveThreshold mechanism. Indeed, the high-level structure of our proof follows that of AboveThreshold: we couple $k$ with $k + 1$ in the two neighboring datasets, and bound the ratio of the output probabilities in the two cases. This is formalized in the proof below.

---

**Algorithm 4** Generalized AboveThreshold Mechanism with Random Dropping.

---

**Parameters:** Distribution $\mathcal{E}$, $\varepsilon_i$-DP mechanisms $\mathcal{M}_i : \mathcal{D} \to \mathcal{O}$, additional privacy budget $\varepsilon' > 0$, and threshold $\tau \in \mathcal{O}$
**Input:** Dataset $D$.
  Sample $k \sim \mathrm{Geom}_{e^{-\varepsilon'}}$
  **for** $i = 1, \ldots, d$ **do**
    Sample $y_i \sim \mathrm{Ber}(e^{-\varepsilon_i \cdot k})$                                    {random drop}
    **if** $y_i = 1$ **then**
      $o_i \leftarrow \mathcal{M}_i(D_i)$
      **if** $o_i \geq \tau$ **then**
        **return** $(o_i, i)$
  **return** $\perp$

---

Our privacy guarantee for Algorithm 4 is stated below. It says that, if the final output is from $\mathcal{M}_i$, then the privacy budget we pay is only $2\varepsilon_i + \varepsilon'$. The value of $\varepsilon'$ can be arbitrarily small, although setting it too small results in a larger drop probability. The latter can be mitigated by repeating each mechanism $\mathcal{M}_i$ multiple times in the input, which allows us to set that the desired expected number of times that each mechanism is run.

**Theorem 14** (Ex-post Generalized AboveThreshold). *Define a function $\tilde{\varepsilon}$ such that $\tilde{\varepsilon}(o, i) = 2\varepsilon_i + \varepsilon'$ and $\tilde{\varepsilon}(\perp) = \varepsilon'$. Then, Algorithm 4 is ex-post $\tilde{\varepsilon}$-DP.*

*Proof of Theorem 14.* Consider neighboring datasets $D \sim D'$. Let $\mathcal{A}, \mathcal{A}'$ be the output distributions of Algorithm 1 on $D, D'$, respectively and let $\mathcal{Q}_i, \mathcal{Q}_i'$ be the output distributions of $\mathcal{M}_i$ on $D, D'$, respectively. Furthermore, let $\mathcal{O}_{\geq \tau} := \{o' \in \mathcal{O} \mid o' \geq \tau\}$.

Consider any output $(o, i) \in \mathcal{O} \times [d]$. Note that, if $o < \tau$, then $\mathcal{A}(o, i) = \mathcal{A}'(o, i) = 0$. Otherwise, if $o \geq \tau$, then the algorithm outputs $(o, i)$ iff $y_j = 0$ or $o_j < \tau$ for all $j < i$, and $y_i = 1$ and $o_i = o$. Thus, we have

$$\mathcal{A}(o, i) = \sum_{k=0}^{\infty} \mathrm{Geom}_{e^{-\varepsilon'}}(k) \cdot e^{-\varepsilon_i k} \mathcal{Q}_i(o) \cdot \prod_{j < i} \left(1 - e^{-\varepsilon_j k} \mathcal{Q}_j(\mathcal{O}_{\geq \tau})\right).$$

Since $\mathcal{M}_j$ is $\varepsilon_j$-DP, it holds that $\mathcal{Q}_j(\mathcal{O}_{\geq \tau}) \geq e^{-\varepsilon_j} \mathcal{Q}_j'(\mathcal{O}_{\geq \tau})$. Similarly, we have $\mathcal{Q}_i(o) \leq e^{\varepsilon_i} \cdot \mathcal{Q}_i'(o)$. Finally, (1) implies that $\mathrm{Geom}_{e^{-\varepsilon'}}(k) \leq e^{\varepsilon'} \cdot \mathrm{Geom}_{e^{-\varepsilon'}}(k + 1)$. Plugging these into the above gives

$$\mathcal{A}(o, i) \leq \sum_{k=0}^{\infty} \left(e^{\varepsilon'} \cdot \mathrm{Geom}_{e^{-\varepsilon'}}(k + 1)\right) \cdot e^{-\varepsilon_i k} \cdot \left(e^{\varepsilon_i} \mathcal{Q}_i'(o)\right) \cdot \prod_{j \neq i} \left(1 - e^{-\varepsilon_j k} \cdot \left(e^{-\varepsilon_j} \mathcal{Q}_j'(\mathcal{O}_{\geq \tau})\right)\right)$$

$$= e^{2\varepsilon_i + \varepsilon'} \sum_{k=0}^{\infty} \mathrm{Geom}_{e^{-\varepsilon'}}(k + 1) \cdot e^{-\varepsilon_i(k+1)} \mathcal{Q}_i'(o) \cdot \prod_{j \neq i} \left(1 - e^{-\varepsilon_j(k+1)} \mathcal{Q}_j'(\mathcal{O}_{\geq \tau})\right)$$

$$\leq e^{2\varepsilon_i + \varepsilon'} \cdot \mathcal{A}'(o, i).$$

Finally, consider the output $\perp$. For the algorithm to output $\perp$, we must have $y_j = 0$ or $o_j < \tau$ for all $j \in [d]$. Similar to above, we thus have

$$\mathcal{A}(\perp) = \sum_{k=0}^{\infty} \mathrm{Geom}_{e^{-\varepsilon'}}(k) \cdot \prod_{j \in [d]} \left(1 - e^{-\varepsilon_j k} \mathcal{Q}_j(\mathcal{O}_{\geq \tau})\right).$$

$$\leq \sum_{k=0}^{\infty} \left(e^{\varepsilon'} \cdot \mathrm{Geom}_{e^{-\varepsilon'}}(k + 1)\right) \cdot \prod_{j \in [d]} \left(1 - e^{-\varepsilon_j k} \cdot \left(e^{-\varepsilon_j} \mathcal{Q}_j'(\mathcal{O}_{\geq \tau})\right)\right).$$

$$= e^{\varepsilon'} \sum_{k=0}^{\infty} \mathrm{Geom}_{e^{-\varepsilon'}}(k + 1) \cdot \prod_{j \in [d]} \left(1 - e^{-\varepsilon_j(k+1)} \mathcal{Q}_j'(\mathcal{O}_{\geq \tau})\right)$$

$$\leq e^{\varepsilon'} \cdot \mathcal{A}'(\perp). \qquad \square$$

Theorem 14 can be viewed as a generalization of Liu and Talwar [2019] who prove a similar statement for ex-ante DP. Nevertheless, we stress that our mechanism is based on a different technique. As demonstrated in the next section, our technique is more robust as it generalizes to hyperparameter tuning (without a known threshold) with a similar privacy guarantee, whereas Liu and Talwar [2019] have to pay a factor of 3 instead of 2 in that setting.

## B   Missing proofs for Ex-post Rényi DP

To prove Theorem 10, we start by collecting some useful facts. The first is the following inequality which is sometimes called the "reverse Hölder's inequality"; we provide the proof for completeness.

**Lemma 15.** *Let $X$ be a random variable and $f, g$ be any functions on $X$. Then, for any $\alpha > 1$, we have*

$$\mathbb{E}[f(X)]^{\alpha} \mathbb{E}[g(X)]^{1-\alpha} \leq \mathbb{E}[f(X)^{\alpha} g(X)^{1-\alpha}].$$

*Proof.* By Hölder's inequality, we have

$$\mathbb{E}[f(X)^\alpha g(X)^{1-\alpha}]^{\frac{1}{\alpha}}\mathbb{E}[g(X)]^{\frac{\alpha-1}{\alpha}} \ge \mathbb{E}[f(X)].$$

Rearranging this yields the claimed inequality. $\square$

**Lemma 16.** *For all $\varepsilon > 0$, $\alpha > 1$, if $a, b \in (0,1)$ are such that $a^{1-\alpha}b^\alpha \le e^{\varepsilon(\alpha-1)}$, then, for all $\ell > 0$,*

$$(1-a)^\alpha \left(1 - e^{-\varepsilon(1+\ell)}b\right)^{1-\alpha} \le \exp\left(e^{-\varepsilon(1+\alpha\ell)}\right).$$

*Proof.* From $1 + x \le e^x$ for all $x \in \mathbb{R}$, the LHS is at most $\exp((\alpha-1)e^{-\varepsilon(1+\ell)}b - \alpha a)$. It is thus sufficient to bound $(\alpha-1)e^{-\varepsilon(1+\ell)}b - \alpha a$.

To do this, observe that the condition $a^{1-\alpha}b^\alpha \le e^{\varepsilon(\alpha-1)}$ implies

$$a \ge e^{-\varepsilon} \cdot b^{\frac{\alpha}{\alpha-1}}. \tag{4}$$

Thus, we may bound the desired term as follows.

$$(\alpha-1)e^{-\varepsilon(1+\ell)}b - \alpha a$$
$$\overset{(4)}{\le} (\alpha-1)e^{-\varepsilon(1+\ell)}b - \alpha \cdot e^{-\varepsilon} \cdot b^{\frac{\alpha}{\alpha-1}}$$
$$= e^{-\varepsilon}\left((\alpha-1)e^{-\varepsilon\ell} - \alpha b^{\frac{1}{\alpha-1}}\right)b$$
$$= e^{-\varepsilon}\left(\frac{\alpha-1}{\alpha}\right)^{\alpha-1}\left(\left((\alpha-1)e^{-\varepsilon\ell} - \alpha b^{\frac{1}{\alpha-1}}\right)^1\left(\frac{\alpha}{\alpha-1}\cdot b^{\frac{1}{\alpha-1}}\right)^{\alpha-1}\right)$$
$$\overset{(\star)}{\le} e^{-\varepsilon}\left(\frac{\alpha-1}{\alpha}\right)^{\alpha-1}\left(\frac{\left((\alpha-1)e^{-\varepsilon\ell} - \alpha b^{\frac{1}{\alpha-1}}\right) + (\alpha-1)\cdot\left(\frac{\alpha}{\alpha-1}\cdot b^{\frac{1}{\alpha-1}}\right)}{\alpha}\right)^\alpha$$
$$= e^{-\varepsilon}\left(\frac{\alpha-1}{\alpha}\right)^{\alpha-1}\left(\frac{(\alpha-1)e^{-\varepsilon\ell}}{\alpha}\right)^\alpha$$
$$= e^{-\varepsilon(1+\alpha\ell)}\left(\frac{\alpha-1}{\alpha}\right)^{2\alpha-1}$$
$$\le e^{-\varepsilon(1+\alpha\ell)},$$

where we use the weighted AM–GM inequality for $(\star)$. $\square$

*Proof of Theorem 10.* We will use the same notations as in the proof of Theorem 8.

First, let us rearrange the term we wish to bound;

$$\sum_{\tilde{o}\in\mathcal{O}\times[d]\cup\{\perp\}}(\mathcal{A}(\tilde{o}))^\alpha(\mathcal{A}'(\tilde{o}))^{1-\alpha}e^{(1-\alpha)\tilde{\varepsilon}(\tilde{o})}$$
$$= (\mathcal{A}(\perp))^\alpha(\mathcal{A}'(\perp))^{1-\alpha}e^{-(\alpha-1)\tilde{\varepsilon}(\perp)} + \sum_{o\in\mathcal{O},i\in[d]}(\mathcal{A}(o,i))^\alpha(\mathcal{A}'(o,i))^{1-\alpha}e^{-(\alpha-1)\tilde{\varepsilon}(o,i)}$$
$$\le \frac{1}{\tau+1} + \sum_{o\in\mathcal{O},i\in[d]}(\mathcal{A}(o,i))^\alpha(\mathcal{A}'(o,i))^{1-\alpha}e^{-(\alpha-1)\tilde{\varepsilon}(o,i)}. \tag{5}$$

We will now bound each term $(\mathcal{A}(o,i))^\alpha(\mathcal{A}'(o,i))^{1-\alpha}$ above. We have

$$\mathcal{A}(o,i) = \mathbb{E}_{x\sim\mathrm{Exp}_{\varepsilon'}}\left[e^{-\varepsilon_i\cdot x}\mathcal{Q}_i(o)\prod_{j\in[d]\setminus\{i\}}(1 - e^{-\varepsilon_j\cdot x}\mathcal{Q}_j(U_{o,i}^j))\right]$$
$$= \mathcal{Q}_i(o)\cdot\mathbb{E}_{x\sim\mathrm{Exp}_{\varepsilon'}}\left[e^{-\varepsilon_i\cdot x}\prod_{j\in[d]\setminus\{i\}}(1 - e^{-\varepsilon_j\cdot x}\mathcal{Q}_j(U_{o,i}^j))\right],$$

and

$$\mathcal{A}'(o,i)$$

$$= \mathbb{E}_{x \sim \mathrm{Exp}_{\varepsilon'}} \left[ e^{-\varepsilon_i \cdot x} \mathcal{Q}'_i(o) \prod_{j \in [d] \smallsetminus \{i\}} (1 - e^{-\varepsilon_j \cdot x} \mathcal{Q}'_j(U^j_{o,i})) \right]$$

$$= \int_0^\infty \varepsilon' e^{-\varepsilon' x} \cdot \left( e^{-\varepsilon_i \cdot x} \mathcal{Q}'_i(o) \right) \prod_{j \in [d] \smallsetminus \{i\}} (1 - e^{-\varepsilon_j \cdot x} \mathcal{Q}'_j(U^j_{o,i})) \, dx$$

$$\geq \int_{1+\ell_i}^\infty \varepsilon' e^{-\varepsilon' x} \cdot \left( e^{-\varepsilon_i \cdot x} \mathcal{Q}'_i(o) \right) \prod_{j \in [d] \smallsetminus \{i\}} (1 - e^{-\varepsilon_j \cdot x} \mathcal{Q}'_j(U^j_{o,i})) \, dx$$

$$= e^{-(1+\ell_i)(\varepsilon'+\varepsilon_i)} \mathcal{Q}'_i(o) \cdot \int_0^\infty \varepsilon' e^{-\varepsilon' x} \cdot e^{-\varepsilon_i \cdot x} \prod_{j \in [d] \smallsetminus \{i\}} (1 - e^{-\varepsilon_j(1+\ell_i)} \cdot e^{-\varepsilon_j \cdot x} \mathcal{Q}'_j(U^j_{o,i})) \, dx$$

$$= e^{-(1+\ell_i)(\varepsilon'+\varepsilon_i)} \mathcal{Q}'_i(o) \cdot \mathbb{E}_{x \sim \mathrm{Exp}_{\varepsilon'}} \left[ e^{-\varepsilon_i \cdot x} \prod_{j \in [d] \smallsetminus \{i\}} (1 - e^{-\varepsilon_j(1+\ell_i)} \cdot e^{-\varepsilon_j \cdot x} \mathcal{Q}'_j(U^j_{o,i})) \right],$$

where the integrals are due to exponential random variables $x$.

Combining these two inequalities together with Lemma 15, we get that

$$(\mathcal{A}(o,i))^\alpha (\mathcal{A}'(o,i))^{1-\alpha}$$
$$\leq e^{(\alpha-1)(1+\ell_i)(\varepsilon'+\varepsilon_i)} (\mathcal{Q}_i(o))^\alpha (\mathcal{Q}'_i(o))^{1-\alpha}$$

$$\cdot \mathbb{E}_{x \sim \mathrm{Exp}_{\varepsilon'}} \left[ e^{-\varepsilon_i \cdot x} \cdot \prod_{j \in [d] \smallsetminus \{i\}} (1 - e^{-\varepsilon_i \cdot x} \mathcal{Q}_j(U^j_{o,i}))^\alpha (1 - e^{-\varepsilon_i(1+\ell_i)} \cdot e^{-\varepsilon_i \cdot x} \mathcal{Q}'_j(U^j_{o,i}))^{1-\alpha} \right].$$

To bound the inner term, first consider a post-processing of mechanism $M_j$ where, after running, we only output whether the score is greater than $s_i$. Since this is a post-processing of $M_j$, this mechanism is also $(\alpha, \varepsilon_j)$-RDP. As such, we have $(\mathcal{Q}_j(U^j_{o,i}))^{1-\alpha}(\mathcal{Q}'_j(U^j_{o,i}))^\alpha \leq e^{\varepsilon_j(\alpha-1)}$. Thus, we may apply Lemma 16 to conclude that

$$(1 - e^{-\varepsilon_i \cdot x} \mathcal{Q}_j(U^j_{o,i}))^\alpha (1 - e^{-\varepsilon_i(1+\ell_i)} \cdot e^{-\varepsilon_i \cdot x} \mathcal{Q}'_j(U^j_{o,i}))^{1-\alpha} \leq \exp\left( e^{-\varepsilon_j(1+\alpha\ell_i)} \right).$$

Plugging this into the above, we arrive at

$$(\mathcal{A}(o,i))^\alpha (\mathcal{A}'(o,i))^{1-\alpha}$$

$$\leq e^{(\alpha-1)(1+\ell)(\varepsilon'+\varepsilon_i)} (\mathcal{Q}_i(o))^\alpha (\mathcal{Q}'_i(o))^{1-\alpha} \cdot \exp\left( \sum_{j \in [d] \smallsetminus \{i\}} e^{-\varepsilon_j(1+\alpha\ell_i)} \right) \mathbb{E}_{x \sim \mathrm{Exp}_{\varepsilon'}} \left[ e^{-\varepsilon_i \cdot x} \right]$$

$$= e^{(\alpha-1)(1+\ell)(\varepsilon'+\varepsilon_i)} (\mathcal{Q}_i(o))^\alpha (\mathcal{Q}'_i(o))^{1-\alpha} \cdot \exp\left( \sum_{j \in [d] \smallsetminus \{i\}} e^{-\varepsilon_j(1+\alpha\ell_i)} \right) \tau_i$$

$$\leq \frac{\tau_i}{\tau+1} \cdot e^{(\alpha-1)(\tilde{\varepsilon}(o,i)-\varepsilon_i)} (\mathcal{Q}_i(o))^\alpha (\mathcal{Q}'_i(o))^{1-\alpha},$$

where in the last inequality we use our choice of $\ell_i$ and $\tilde{\varepsilon}(o,i)$.

Combining with (5), we thus get

$$\sum_{\tilde{o} \in \mathcal{O} \times [d] \cup \{\perp\}} (\mathcal{A}(\tilde{o}))^\alpha (\mathcal{A}'(\tilde{o}))^{1-\alpha} e^{(1-\alpha)\tilde{\varepsilon}(\tilde{o})}$$

$$\leq \frac{1}{\tau+1} + \sum_{i \in [d]} \sum_{o \in \mathcal{O}} \frac{\tau_i}{\tau+1} e^{-(\alpha-1)\varepsilon_i} \frac{(\mathcal{Q}_i(o))^\alpha}{(\mathcal{Q}'_i(o))^{\alpha-1}}$$

$$= \frac{1}{\tau+1} + \sum_{i \in [d]} \frac{\tau_i}{\tau+1} e^{-(\alpha-1)\varepsilon_i} \left( \sum_{o \in \mathcal{O}} \frac{(\mathcal{Q}_i(o))^\alpha}{(\mathcal{Q}'_i(o))^{\alpha-1}} \right)$$

$$\leq \frac{1}{\tau+1} + \sum_{i\in[d]} \frac{\tau_i}{\tau+1} \cdot e^{-(\alpha-1)\varepsilon_i} e^{(\alpha-1)\varepsilon_i}$$

$$= \frac{1}{\tau+1} + \sum_{i\in[d]} \frac{\tau_i}{\tau+1} = 1. \qquad \square$$

## C  Fully-Adaptive Composition with Ex-Post Rényi DP

*Proof of Theorem 11.* Without loss of generality, we can assume that the adversary is always issuing queries such that $\sum_{j=1}^{i-1} \tilde{\varepsilon}_j(o_j) + \sup_o \tilde{\varepsilon}_i(o_i) \leq \varepsilon$ for all $i \in [n]$. Let us denote the query issued by the adversary after seeing $o_1,\ldots, o_{i-1}$ as $D^{(0)}_{o_1,\ldots,o_{i-1}}, D^{(1)}_{o_1,\ldots,o_{i-1}}, \tilde{\varepsilon}_{o_1,\ldots,o_{i-1}}$, and $\mathcal{M}_{o_1,\ldots,o_{i-1}}$. Let us also denote the distribution of $\mathcal{M}_{o_1,\ldots,o_{i-1}}(D^b_{o_1,\ldots,o_{i-1}})$ by $P^{(b)}_{o_1,\ldots,o_{i-1}}$. Note that

$$e^{(\alpha-1)\mathrm{D}_\alpha\left(\mathrm{IT}^0(\mathcal{F}_{\alpha,\varepsilon};\mathcal{A}) \,\|\, \mathrm{IT}^1(\mathcal{F}_{\alpha,\varepsilon};\mathcal{A})\right)}$$

$$= \sum_{o_1,o_2,\ldots,o_n} \frac{(P^{(0)}(o_1)P^{(0)}_{o_1}(o_2)\cdots P^{(0)}_{o_1,\ldots,o_{n-1}}(o_n))^\alpha}{(P^{(1)}(o_1)P^{(1)}_{o_1}(o_2)\cdots P^{(1)}_{o_1,\ldots,o_{n-1}}(o_n))^{\alpha-1}}$$

$$= \sum_{o_1,o_2,\ldots,o_{n-1}} \frac{(P^{(0)}(o_1)P^{(0)}_{o_1}(o_2)\cdots P^{(0)}_{o_1,\ldots,o_{n-2}}(o_{n-1}))^\alpha}{(P^{(1)}(o_1)P^{(1)}_{o_1}(o_2)\cdots P^{(1)}_{o_1,\ldots,o_{n-2}}(o_{n-1}))^{\alpha-1}} \left( \sum_{o_n} \frac{(P^{(0)}_{o_1,\ldots,o_{n-1}}(o_n))^\alpha}{(P^{(1)}_{o_1,\ldots,o_{n-1}}(o_n))^{\alpha-1}} \right).$$

Further, note that $\mathcal{M}_{o_1,\ldots,o_{n-1}}$ is $(\alpha, \tilde{\varepsilon}_{o_1,\ldots,o_{n-1}})$-RDP and hence,

$$\sum_{o_n} \frac{(P^{(0)}_{o_1,\ldots,o_{n-1}}(o_n))^\alpha}{(e^{\tilde{\varepsilon}_{o_1,\ldots,o_{n-1}}(o_n)} P^{(1)}_{o_1,\ldots,o_{n-1}}(o_n))^{\alpha-1}} \leq 1.$$

Let $L(o_1,\ldots,o_k) = \dfrac{(P^{(0)}(o_1)P^{(0)}_{o_1}(o_2)\cdots P^{(0)}_{o_1,\ldots,o_{k-1}}(o_k))^\alpha}{(e^{\tilde{\varepsilon}(o_1)}P^{(1)}(o_1)e^{\tilde{\varepsilon}_{o_1}(o_2)}P^{(1)}_{o_1}(o_2)\cdots e^{\tilde{\varepsilon}_{o_1,\ldots,o_{k-1}}(o_k)}P^{(1)}_{o_1,\ldots,o_{k-1}}(o_k))^{\alpha-1}}$. Then

$$\frac{e^{(\alpha-1)\mathrm{D}_\alpha\left(\mathrm{IT}^0(\mathcal{F}_{\alpha,\varepsilon};\mathcal{A}) \,\|\, \mathrm{IT}^1(\mathcal{F}_{\alpha,\varepsilon};\mathcal{A})\right)}}{e^{(\alpha-1)\varepsilon}}$$

$$= \frac{1}{e^{(\alpha-1)\varepsilon}} \sum_{o_1,o_2,\ldots,o_n} \frac{(P^{(0)}(o_1)P^{(0)}_{o_1}(o_2)\cdots P^{(0)}_{o_1,\ldots,o_{n-1}}(o_n))^\alpha}{(P^{(1)}(o_1)P^{(1)}_{o_1}(o_2)\cdots P^{(1)}_{o_1,\ldots,o_{n-1}}(o_n))^{\alpha-1}}$$

$$\leq \sum_{o_1,o_2,\ldots,o_n} \frac{(P^{(0)}(o_1)P^{(0)}_{o_1}(o_2)\ldots P^{(0)}_{o_1,\cdots,o_{n-1}}(o_n))^\alpha}{(e^{\tilde{\varepsilon}(o_1)}P^{(1)}(o_1)e^{\tilde{\varepsilon}_{o_1}(o_2)}P^{(1)}_{o_1}(o_2)\cdots e^{\tilde{\varepsilon}_{o_1,\ldots,o_{n-1}}(o_n)}P^{(1)}_{o_1,\ldots,o_{n-1}}(o_n))^{\alpha-1}}$$

$$= \sum_{o_1,\ldots,o_n} L(o_1,\ldots,o_n)$$

$$= \sum_{o_1,\ldots,o_{n-1}} L(o_1,\ldots,o_{n-1}) \left( \sum_{o_n} \frac{(P^{(0)}_{o_1,\ldots,o_{n-1}}(o_n))^\alpha}{(e^{\tilde{\varepsilon}_{o_1,\ldots,o_{n-1}}(o_n)} P^{(1)}_{o_1,\ldots,o_{n-1}}(o_n))^{\alpha-1}} \right)$$

$$\leq \sum_{o_1,\ldots,o_{n-1}} L(o_1,\ldots,o_{n-1})$$

$$\leq \sum_{o_1,\ldots,o_{n-2}} L(o_1,\ldots,o_{n-2}) \left( \sum_{o_{n-1}} \frac{(P^{(0)}_{o_1,\ldots,o_{n-2}}(o_{n-1}))^\alpha}{(e^{\tilde{\varepsilon}_{o_1,\ldots,o_{n-2}}(o_{n-1})} P^{(1)}_{o_1,\ldots,o_{n-2}}(o_{n-1}))^{\alpha-1}} \right)$$

$$\vdots$$

$$\leq \sum_{o_1} L(o_1) \leq 1, \text{ which implies that } \mathrm{D}_\alpha\left(\mathrm{IT}^0(\mathcal{F}_{\alpha,\varepsilon};\mathcal{A}) \,\|\, \mathrm{IT}^1(\mathcal{F}_{\alpha,\varepsilon};\mathcal{A})\right) \leq \varepsilon. \qquad \square$$

