# OpenReview forum: "Private Hyperparameter Tuning with Ex-Post Guarantee"
_NeurIPS.cc/2025/Conference — NeurIPS 2025 spotlight_

### Official Review · Reviewer_AAhR · 2025-06-26

**Clarity:** 2
**Significance:** 3
**Originality:** 2
**Rating:** 4
**Confidence:** 3

**Summary:**

The paper presents a novel framework for hyperparameter tuning in differential privacy (DP) with a utility-first perspective, building on the ex-post DP concept introduced by Wu et al. (2019). It proposes a general algorithm that supports any sequence of private estimators, achieving ex-post DP and Rényi DP guarantees with at most a doubling of the privacy budget. The algorithm extends to complex mechanisms like DP-SGD, unlike prior work limited to Laplace or Gaussian noise. Additionally, the paper introduces a privacy filter for fully adaptive composition in ex-post Rényi DP and provides empirical evidence of superior performance in machine learning tasks, such as linear regression on Twitter data and classification on MNIST.

**Questions:**

1. Can the authors clarify or correct the inconsistency between Definition 2 and Definition 5 regarding the summation over the output set?
2. Could the authors add a note near Algorithm 1 or in its description to highlight the unconventional use of the probability of failurefor the geometric distribution? This would prevent implementation errors and could improve the Clarity score to 3.
3. Can the authors discuss the implications of the uniform $\varepsilon' $ term in Theorem 8, which reduces the output-specific flexibility of ex-post DP?
4. In Theorem 10, are the parameters $l_1, \dots, l_d $ arbitrary, or do they require optimization? If optimization is not needed, would swapping the quantifier order improve clarity?
5. Can the authors provide a better introduction or motivation for Section 4 to connect it to the paper’s main narrative?

**Ethical Concerns:**

["NO or VERY MINOR ethics concerns only"]

**Final Justification:**

After reading the author’s response, I’ve decided to keep my positive rating of the article.

**Limitations:**

Yes

**Quality:**

3

**Strengths And Weaknesses:**

Strengths
- The algorithm significantly generalizes prior work by Wu et al. (2019) and Liu and Talwar (2019), extending ex-post DP to arbitrary private estimators, including DP-SGD.
- The proof of Theorem 8, which establishes ex-post pure-DP guarantees, is notably simple and leverages the Sparse Vector Technique. Its elementary nature enhances its accessibility and strengthens the theoretical contribution.
- The experimental results in Section 5.2 demonstrate substantial improvements in machine learning tasks.

Weaknesses
- There is an inconsistency between Definition 2 (ex-post pure-DP) and Definition 5 (ex-post approximate-DP), as Definition 5 includes a summation over the output set, absent in Definition 2.
- The parametrization of the geometric distribution in Algorithm 1, using the probability of failure instead of the conventional probability of success, is unconventional and not highlighted. Without reading the “Probability Notation” section, readers might implement the algorithm incorrectly.
- Theorem 8 introduces a uniform additive term across all indices, which partially undermines the output-specific flexibility of ex-post DP, aligning it closer to ex-ante DP. This trade-off is not sufficiently discussed.
- A typo in the proof of Theorem 8 (final line) omits the $\varepsilon$-term and incorrectly uses $\mathcal{A} $ instead of $\mathcal{A}' $, which could compromise the proof’s integrity.
- Section 4 is poorly introduced and feels disconnected from the rest of the paper, lacking a clear transition or motivation.
- The general writing could be improved, with occasional unclear phrasing and lack of coherence in some sections, particularly Section 4.

---

> ### Author Rebuttal · Authors · 2025-07-30
>
> Thank you for your detailed review and suggestions.
>
>
> # Response to Weaknesses
>
>
> * Regarding the last expression of the proof of Theorem 8: We thank the reviewer for pointing this out. Indeed this is a typo and should be changed to $e^{\\epsilon_i + \\epsilon'} \\cdot \\mathcal{A}'(o, i)$. We have rechecked the proof and can confirm that, apart from this typo, the proof is correct.
>
>
> # Response to Questions
>
>
> 1. We wish to remark that the two versions (with and without the sum) of the definitions are equivalent for pure-DP  (i.e., $\\delta = 0$). One direction (with the sum to without the sum) is obvious by taking $\\mathcal{O} = \\{o\\}$. The other direction can be seen by taking the sum of the inequality over all $o \\in \\mathcal{O}$. We also note that both versions are standard in DP literature; indeed the original DP paper (Dwork et al., 2006) has the definition of pure-DP without the sum, in a similar manner as our Definition 2.
>
>
> 2. Thank you for pointing this out. We will make sure to change this and use $p$ as the success probability in the revised version.
>
>
> 3. This term is actually not a problem since we can choose $\\epsilon’$ to be arbitrarily small, and thus the privacy overhead from this additive term can be made negligible. Note that, to retain similar utility after decreasing $\\epsilon’$, we can simply duplicate the mechanisms so that each mechanism is run multiple times (see Line 178-191); this would result in the same utility and lower privacy at the expense of computational cost. (Note that a similar tradeoff also occurs in ex-ante DP hyperparameter tuning algorithm of Papernot–Steinke; see their Corollary 3.)
>
>
> 4. Note that $\\ell_i$’s are *not* data dependent so they can be set entirely based on the $\\epsilon_i$’s values; i.e., one may choose $\\ell_i$ that minimizes $(2 + \\ell_i) \\epsilon_i + (1 + \\ell_i) \epsilon’ + \\frac{\\sum_{j \neq i} e^{-\\epsilon_j (1 + \\alpha  \\ell_i)}}{\alpha - 1}$ (note that it doesn’t depend on $\\ell_j$ for $j \\neq i$ or the dataset).
>
>
> 5. As alluded to in the paragraph starting at line 92, privacy filter is the typical approach to convert ex-post guarantee into the standard ex-ante guarantee. Here one would have a target ex-ante privacy budget (denoted by $\\epsilon$ in Algorithm 2) and the privacy filter allows us to run multiple ex-post DP subroutines. At the end, the guarantee of the entire algorithm is *ex-ante*. In other words, this can be thought of as a “composition theorem” for privacy, except that it allows ex-post DP subroutines and yet yields the more traditional final ex-ante DP. Finally, we note that we focus on RDP in Section 4 since a natural privacy filter for e.g. pure-DP is already known (Lebensold et al., 2024).

---

### Official Review · Reviewer_ULcx · 2025-06-27

**Clarity:** 3
**Significance:** 3
**Originality:** 3
**Rating:** 5
**Confidence:** 3

**Summary:**

Wu et al. 2019 proposed the framework of Ex-post DP, which aims to search for the smallest privacy budget that passes  a specified  utility.  Despite practical relevance, literature on Ex-post is limited and existing methods do not apply to DP-SGD.

Author’s contribution starts by extending ex-post DP to ex-post RDP. Their main contribution is a new algorithm and a bound (inspired from the proof of SVT) for DP selection problem under ex-post DP/RDP.  The DP selection problem has applications in hyperparameter tuning/model selection process.

A major limitation of seminal results from Liu and Talwar 2019 and Papernot and Steinke 2022 for (ex-ante) DP-aware hyperparameter tuning (model selection) is that they assume the same privacy cost for all base mechanisms. This means one can tune the batch size, DP noise scales or number of epochs for a fixed epsilon, but not the base epsilon itself.

Take-away from their bound: Among several candidate ex-ante DP mechanisms with (possibly) heterogeneous ex-ante $\epsilon$’s, if the ith mechanism produces the best result, then the total ex-post budget of the entire exploration is only $2\epsilon_i$. The factor of 2 seems unavoidable.

They also propose a privacy filter to adaptively compose several ex-post RDP mechanisms to obtain the final ex-ante guarantee.

**Questions:**

1)	Algorithm 1 does not include the step for checking whether output passes the utility threshold for each $M_i$. Can you please clarify?

2)	The line 138 seems to suggest that 'M_1,...,M_d' can be executed multiple times. However, there is no outer loop in Algorithm 1. So, it seems the number of distinct mechanisms need to be much lower than d to have each mechanism run multiple times. If my understanding is correct, please mention this explicitly.

**Ethical Concerns:**

["NO or VERY MINOR ethics concerns only"]

**Final Justification:**

My issues have been resolved.

Upon acceptance, authors should consider including their responses for my questions above.
This is a strong paper and meets NeurlPS bar, IMO.

**Limitations:**

Yes

**Quality:**

3

**Strengths And Weaknesses:**

Strengths:

1)	You only incur the privacy cost of  the best mechanism which may have much smaller budget than the suboptimal mechanisms with higher budgets. This effect is quite surprising.
2)	The bound places no restrictions on the mechanisms used, and also applies to DP-SGD which is the main the main beneficiary of hyperparameter tuning methods.

Weakness:
1)	The runtime of this method has higher variance than Papernot & Steinke for the same expectation. But authors already mention this.
2)	The final ex-post epsilon can vary a lot, e.g., when the accuracy gap between the best and second-best mechanism is tiny but their base epsilons differ a lot.
3)	The bound introduces several new hyperparameters ($\ell_i’s $). These can be set equal, and upper bounded, but it’s not clear how to choose these in general.

---

> ### Author Rebuttal · Authors · 2025-07-30
>
> Thank you for your detailed review and suggestions.
>
>
> # Response to Weaknesses
>
>
> Note that $\\ell_i$’s are *not* data dependent so they can be set entirely based on the $\\epsilon_i$’s values; i.e., one may choose $\\ell_i$ that minimizes $(2 + \\ell_i) \\epsilon_i + (1 + \\ell_i) \epsilon’ + \\frac{\\sum_{j \neq i} e^{-\\epsilon_j (1 + \\alpha  \\ell_i)}}{\alpha - 1}$ (note that it doesn’t depend on $\\ell_j$ for $j \\neq i$ or the dataset).
>
>
> # Response to Questions
>
>
> 1. Indeed, one of the strengths of our algorithm is that it does not need an a priori threshold since it simply outputs the best response with the highest “score” (based on the ordering of $\\mathcal{O}$ that we can choose). However, if one insists on using a threshold, then our algorithm can be easily adapted: We can assign every output below the threshold with “score” of $-\\infty$ (and every output above the threshold a positive score)  and add another 0-DP mechanism $\\mathcal{M}_{d + 1}$ that always output a dummy output with score 0. In this way, we ensure that we only output an element that exceeds the threshold.
> 2. We apologize for the confusing phrases that we used. You are correct that in our Algorithm 1 description each mechanism is run exactly once. However, as explained in the subsequent paragraphs (Line 178-192), we can run each mechanism more than once by a simple “repetition trick” where we duplicate each mechanism multiple times in the input sequence. Note that, in terms of privacy, there is no cost to such repetition trick since the guarantee in Theorem 8 does *not* depend on the sequence length.

---

> > ### Comment · Reviewer_ULcx · 2025-08-04
> >
> > Thanks for your response. I will keep my positive score. Good luck.

---

### Official Review · Reviewer_gugu · 2025-07-01

**Clarity:** 2
**Significance:** 3
**Originality:** 3
**Rating:** 5
**Confidence:** 3

**Summary:**

This work introduces a hyperparameter tuning algorithm that provides ex-post DP guarantees, allowing a utility-first approach to DP selection. They propose a privacy filter that allows using their ex-post Rényi DP algorithm interactively ensuring a ex-ante guarantee. The authors demonstrate that they can achieve stricter privacy guarantees than prior work for ML applications.

**Questions:**

Some clarification is needed for Theorem 11—for example, the notation $IT^0$ and $IT^1$ is used without definition.

Minor suggestions and comments:
- shortly mention the definition of privacy filter
- briefly discuss what having a larger variance implies (Figure 1)
- Definition 6: is there a reason why you use $\epsilon$ instead of $\tilde{\epsilon}$ here?
- Line 93: missing brackets (Theorem 10)
- for better reproducibility, mention the CNN model architecture (in 5.2 second experiment)

**Ethical Concerns:**

["NO or VERY MINOR ethics concerns only"]

**Final Justification:**

I believe the paper is technically solid and, while not being groundbreaking, addresses a relevant problem in the field. The authors have also adequately addressed the reviewers' comments.

**Limitations:**

yes

**Quality:**

3

**Strengths And Weaknesses:**

This paper is well-motivated and addresses a relevant problem within the field. The authors provide a thorough theoretical treatment of their novel algorithm (hyperparameter tuning via random dropping) proving that it guarantees pure DP and Rényi DP respectively. They also evaluate their algorithm empirically, confirming its practical benefits (and limitations) on two types of problems (analytical and ML).

---

> ### Author Rebuttal · Authors · 2025-07-30
>
> Thank you for your detailed review and suggestions.
>
>
> # Response to Questions
>
>
> As alluded to in the paragraph starting at line 92, privacy filter is the typical way to build a system where an adversary is allowed to run ex-ante private subroutines interactively and the filter halts the computation if a target ex-ante privacy guarantee would be violated by request.
>
>
> However, our paper considers a generalization of this concept to a setting where the subroutines are allowed to be *ex-post* private. In other words, the filter we consider allows converting ex-post guarantee into the standard ex-ante guarantee. Here one would have a target ex-ante privacy budget (denoted by $\\epsilon$ in Algorithm 2) and the privacy filter allows us to run multiple ex-post DP subroutines. At the end, the guarantee of the entire algorithm is *ex-ante*. In other words, this can be thought of as a “composition theorem” for privacy, except that it allows ex-post DP subroutines and yet yields the more traditional final ex-ante DP. Finally, we note that we focus on RDP in Section 4 since a natural privacy filter for,  e.g., pure-DP is already known (Lebensold et al., 2024).
>
>
> As for $IT^0$ and $ IT^1$, we define them to be the output of Algorithm 2 for $b = 0$ and $b = 1$, respectively. ($IT$ here stands for “interactive transcript”.)  We will make sure to include the definition more clearly in the main body in the revised version. Thank you for pointing this out.
>
>
> # Response to Comments
>
>
> Re variance and Figure 1: Larger variance in the number of invocation of the mechanisms also implies a larger variance in the running time and a larger “variance” in the utility guarantee. This is because the number of invocations of the Papernot–Steinke mechanism is more well concentrated around the expectation, so their running time and utility are also more well concentrated. (This is perhaps the price we pay for the generality of our algorithm–which works for mechanisms with different DP guarantees–compared to their algorithm.)
>
>
> Re CNN architecture:
> We didn’t optimize the architecture for MNIST (since it was already outperforming linear models) and used the one from Opacus tutorial: it has two convolution layers and two linear layers with max-pooling (with kernel 2 and stride 1) and ReLU after convolution layers and ReLU after linear layers.
> 1. The first convolution layer has out channel 16, kernel 8, and stride 2.
> 2. The second convolution layer has out channel 32, kernel 4, and stride 2.
> 3. The first linear layer has 32 output features.
> 4. The second linear layer has 10 output features for each class.
>
> We will make sure to add the architecture to the final text too.
>
>
> Thank you so much as well for pointing out the other typos.

---

> > ### Comment · Reviewer_gugu · 2025-08-06
> >
> > Thank you for your response. My questions/comments have been addressed and I will keep my positive score.

---

### Official Review · Reviewer_Rabk · 2025-07-03

**Clarity:** 3
**Significance:** 3
**Originality:** 3
**Rating:** 5
**Confidence:** 3

**Summary:**

The paper presents a general algorithm for hyperparameter tuning under ex-post differential privacy, compatible with both pure DP and Rényi DP. It privately selects among multiple candidate mechanisms, incurring privacy cost only for the selected one. The method supports complex workflows like DP-SGD and introduces an ex-post RDP filter to enable interactive composition. Empirical results demonstrate improved privacy-utility tradeoffs compared to prior approaches.

**Questions:**

1. Would the expression above line 178 be more accurately written as $\tilde{\varepsilon}(o, i)$?

2. Can the authors provide more intuition on why the final ex-post DP guarantee depends only on the selected mechanism?

**Ethical Concerns:**

["NO or VERY MINOR ethics concerns only"]

**Final Justification:**

This is a good paper. I will keep my positive score.

**Limitations:**

yes

**Paper Formatting Concerns:**

No formatting concerns as far as I know.

**Quality:**

4

**Strengths And Weaknesses:**

**Strength**:

This work presents a novel result in private hyperparameter tuning, with potential for practical applications. The algorithm’s design is elegant, and the accompanying proof is communicated with clarity. The author also derives a corresponding privacy filter, which appears to be of additional interest.


**Weakness**:

Please refer to the question section

---

> ### Author Rebuttal · Authors · 2025-07-30
>
> Thank you for your detailed review and suggestions.
>
>
> # Response to Questions
>
>
> 1. We thank the reviewer for pointing this out. Indeed this is a typo and should be changed to $e^{\\epsilon_i + \\epsilon'} \\cdot \\mathcal{A}'(o, i)$. We have rechecked the proof and can confirm that, apart from this typo, the proof is correct.
>
> 2. We find this surprising as well! Our rough intuition is that, since we randomly decide if each mechanism’s result is included in the final computation, this allows mitigating the leak of each individual mechanism. However, as can be seen in our paper (and previous works of Liu–Talwar and Papernot–Steinke), this requires a very careful strategy of how each mechanism is probabilistically included, together with a subtle privacy proof.

---

> > ### Comment · Reviewer_Rabk · 2025-08-04
> >
> > Thanks for answering my questions! I will keep my positive score.

---

### Decision · Program_Chairs · 2025-09-17

**Decision:**

Accept (spotlight)

**Comment:**

The reviews agree that this is an interesting, useful and well-executed paper, and would be a useful addition to NeurIPS. Please revisit the discussion with the reviewers while preparing the camera ready version, and try to incorporate the promised changes (especially for the sections where the writing needs to be improved).